METHODS AND RESOURCES

# gen3sis: A general engine for eco-evolutionary simulations of the processes that shape Earth's biodiversity

Oskar Hagen[1,2]*, Benjamin Flück[1,2], Fabian Fopp[1,2], Juliano S. Cabral[3], Florian Hartig[4], Mikael Pontarp[5], Thiago F. Rangel[6], Loïc Pellissier[1,2]

1 Landscape Ecology, Institute of Terrestrial Ecosystems, Department of Environmental Systems Science, ETH Zürich, Zürich, Switzerland, 2 Land Change Science Research Unit, Swiss Federal Institute for Forest, Snow and Landscape Research, WSL, Birmensdorf, Switzerland, 3 Ecosystem Modeling, Center for Computational and Theoretical Biology, University of Würzburg, Würzburg, Germany, 4 Theoretical Ecology, University of Regensburg, Regensburg, Germany, 5 Department of Biology, Lund University, Lund, Sweden, 6 Department of Ecology, Institute of Biological Sciences, Federal University of Goiás, Goiânia, Brazil

* oskar@hagen.bio

**Data Availability Statement:** Gen3sis is implemented in a mix of R and C++ code, and wrapped into an R-package. All high-level functions that the user may interact with are written in R, and

## Abstract

Understanding the origins of biodiversity has been an aspiration since the days of early naturalists. The immense complexity of ecological, evolutionary, and spatial processes, however, has made this goal elusive to this day. Computer models serve progress in many scientific fields, but in the fields of macroecology and macroevolution, eco-evolutionary models are comparatively less developed. We present a general, spatially explicit, eco-evolutionary engine with a modular implementation that enables the modeling of multiple macroecological and macroevolutionary processes and feedbacks across representative spatiotemporally dynamic landscapes. Modeled processes can include species' abiotic tolerances, biotic interactions, dispersal, speciation, and evolution of ecological traits. Commonly observed biodiversity patterns, such as α, β, and γ diversity, species ranges, ecological traits, and phylogenies, emerge as simulations proceed. As an illustration, we examine alternative hypotheses expected to have shaped the latitudinal diversity gradient (LDG) during the Earth's Cenozoic era. Our exploratory simulations simultaneously produce multiple realistic biodiversity patterns, such as the LDG, current species richness, and range size frequencies, as well as phylogenetic metrics. The model engine is open source and available as an R package, enabling future exploration of various landscapes and biological processes, while outputs can be linked with a variety of empirical biodiversity patterns. This work represents a key toward a numeric, interdisciplinary, and mechanistic understanding of the physical and biological processes that shape Earth's biodiversity.

## Introduction

Ecological and evolutionary processes have created various patterns of diversity in living organisms across the globe [1]. Species richness varies across regions, such as continents [2,3], and along spatial and environmental gradients [4,5], such as latitude [6,7]. These well-known

are documented via the standard R / Roxygen help files for R-packages. Runtime-critical functions are implemented in C++ and coupled to R via the Rcpp framework. Additionally, the package provides several convenience functions to generate input data, configuration files and plots, as well as tutorials in the form of vignettes that illustrate how to declare models and run simulations. The software, under an open and free GPL3 license, can be downloaded from CRAN at https://CRAN.R-project.org/package=gen3sis. The development version, open to issue reporting and feature suggestions, is available at https://github.com/project-Gen3sis/R-package. Supporting information, such as notes, scripts, data, figures and animations, are available at https://zenodo.org/record/5006413, facilitating full reproducibility.

**Funding:** OH, JC, FH, MP, TR and LP were part of the sELDiG working group, which was supported by the Synthesis Centre of the German Centre for Integrative Biodiversity Research, Halle-Jena-Leipzig (DFG FZT 118). LP was supported by the Swiss National Science Foundation grant (N˚ 310030_188550). The funders had no role in study design, data collection and analysis, decision to publish, or preparation of the manuscript.

**Competing interests:** The authors have declared that no competing interests exist.

**Abbreviations:** BIC, Bayesian information criteria; gen3sis, general engine for eco-evolutionary simulations; LDG, latitudinal diversity gradient; nLTT, normalized lineage though time; POM, pattern-oriented modeling; SRD, species range decrease.

patterns, derived from the observed multitude of life forms on Earth, have intrigued naturalists for centuries [1,8,9] and stimulated the formulation of numerous hypotheses to explain their origin (e.g., [1,6,7,10,11–15]). Ecologists and evolutionary biologists have attempted to test and disentangle these hypotheses [16], for example, via models of cladogenesis [17] or correlative spatial analyses [18,19]. However, we are only at the beginning of a mechanistic understanding of the ecological and evolutionary dynamics driving diversity patterns [20–23].

The complexity of interacting ecological, evolutionary, and spatial processes limits our ability to formulate, test, and apply the mechanisms forming biodiversity patterns [24,25]. Additionally, multiple processes act and interact with different relative strengths across spatiotemporal scales [20]. Current research suggests that allopatric [22,23,26] and ecological [24] speciation, dispersal [27], and adaptation [28] all act conjointly in interaction with the environment [29,30], producing observed biodiversity patterns [31]. Comprehensive explanations of the origin and dynamics of biodiversity must therefore consider a large number of biological processes and feedbacks [32], including species' ecological and evolutionary responses to their dynamic abiotic environment, acting on both ecological and evolutionary time scales [20,33]. Consequently, biodiversity patterns can rarely be explained by a single hypothesis, as the expectations of the various contending mechanisms are not clearly asserted [20,34].

A decade ago, a seminal paper by Gotelli and colleagues [35] formulated the goal of developing a "general simulation model for macroecology and macroevolution" (hereafter computer models). Since then, many authors have reiterated this call for a broader use of computer models in biodiversity research [20,36,37], prompting the implementation of several models to explore the emergence of patterns [22,23,38,39]. With computer models, researchers can explore the implications of implemented hypotheses and mechanisms and evaluate whether emerging model outputs are compatible with observations. Several case studies have illustrated the feasibility and usefulness of eco-evolutionary computer models in guiding the interpretation of empirical data [23,26,38,40–46]. Moreover, models have reproduced realistic large-scale biodiversity patterns, such as those along latitude [22,39,47], by considering climate and geological dynamics [23,26,38,44] and those related to population isolation, by considering dispersal ability and geographic distance [22,23,26,38,40–44]. For example, computer models have been used to examine how oceans' paleogeography has influenced biodiversity dynamics in marine ecosystems [26,38,43,47]. Despite these recent studies, there is still scope for developing advanced computer models to shed light on the mechanisms underlying biodiversity patterns. In particular, general models that can accommodate and thus contrast several of the hypotheses listed above have utility in our endeavor to better understand and infer the underpinnings of outstanding biodiversity patterns on Earth.

Macroevolutionary studies have highlighted that patterns emerging from simulations are generally sensitive to the mechanisms implemented and to the landscapes upon which mechanisms act [22,26,38,47]. Systematically comparing and exploring the effects of mechanisms and landscapes, however, is often hindered by the lack of flexibility and idiosyncrasies of existing computer models. Most models implement, and thus test, only a limited set of evolutionary processes and hypotheses. Many models are designed for specific and therefore fixed purposes, with spatial and temporal boundaries, ranging from the global [22,26,44] to continental [23] or regional scale [41,42] and from millions of years [38,41,42,47] to thousands of years [22,23]. Moreover, previous eco-evolutionary population models were developed to test a fixed number of mechanisms [22,26,35,38,39,42,44,46,48–52] without having generality build in by design. The diverse input and output formats and limited code availability [53], as well as the different algorithmic implementations, have reduced generality, accessibility, and compatibility between hitherto available models.

Biological hypotheses and landscapes should be compared within a common and standardized platform with the modularity required for flexible explorations of multiple landscapes and processes [35]. Increased generality is thus a desirable feature of computer models that aim to explore the mechanisms and landscapes that shape biodiversity in dynamic systems such as rivers [54], oceans [38,43], islands [41,42,55], and mountains [56,57], or across gradients such as latitude [20,22,47]. Inspired by the mechanistic implementation in existing models used to understand the formation of biodiversity gradients [22,23,26,38–46], we created a simulation engine that can approximate a variety of biological processes over dynamic landscapes. The model integrates mechanisms including (i) allopatric speciation [26,43,44]; (ii) niche evolution [38,39]; and (iii) competitive interactions [23]. This modeling engine offers the possibility to explore the eco-evolutionary dynamics of lineages under a broad range of biological processes and landscapes within a common framework. Simulated species populations occupy a spatial domain (hereafter site) bounded by a combination of geological, climatic, and ecological factors. The sites occupied by a species define the species' realized geographic range (hereafter species range) [58]. The engine tracks species populations over time, which can change as a result of dynamic environments, as well as species dispersal ability, ecological interactions, local adaptation, and speciation. The initial species range and the criteria for speciation, dispersal, ecological interactions, and trait evolution are adjustable mechanisms, allowing the integration of a wide range of hypotheses within the model. Given the flexibility of modifying both mechanisms and landscapes, the engine offers a general tool and is thus named the "general engine for eco-evolutionary simulations" (hereafter gen3sis). We highlight the potential of gen3sis to infer the underlying processes behind biodiversity patterns, thus tackling a longstanding topic in evolutionary ecology.

As an illustration of the flexibility and utility of gen3sis, we implement 5 biological hypotheses with dynamic and static landscapes proposed to explain the latitudinal diversity gradient (LDG) [20] and other biodiversity patterns. More specifically, we implement and analyze (i) a *null model* without ecological interactions, where all terrestrial sites are suitable for all species; (ii) *time for species accumulation* [59–62]; (iii) *diversification rates*, i.e., depending on temperature [63, 64]; (iv) *ecological limits independent* of temperature and aridity; and (v) *ecological limits dependent* on energetic carrying capacity [65,66]. We use this case study to illustrate how simulation results can be compared with multiple empirical biodiversity data, including empirical distribution and phylogenetic patterns of major tetrapod clades (i.e., mammals, birds, amphibians, and reptiles), to inform us about potential mechanisms underlying patterns.

## Engine principles and scope

Gen3sis is a modeling engine, developed for formalizing and testing multiple hypotheses about the emergence of biodiversity patterns. The engine simulates the consequences of multiple customizable processes and landscapes responsible for the appearance (speciation) and disappearance (extinction) of species over evolutionary time scales. Speciation and extinction emerge from ecological and evolutionary mechanisms dependent on dispersal, species interactions, trait evolution, and geographic isolation processes. Customizable eco-evolutionary processes, which interact with dynamic landscapes, make it possible to adjust for various macro-eco-evolutionary hypotheses about specific taxonomic groups, ecosystem types, or processes. We made the engine openly available to the research community in an R package to catalyze an interdisciplinary exploration, application, and quantification of the mechanisms behind biodiversity dynamics. The R statistical programming language and environment [67] is widely used for reproducible and open-source research [68,69], and since its origins, it has

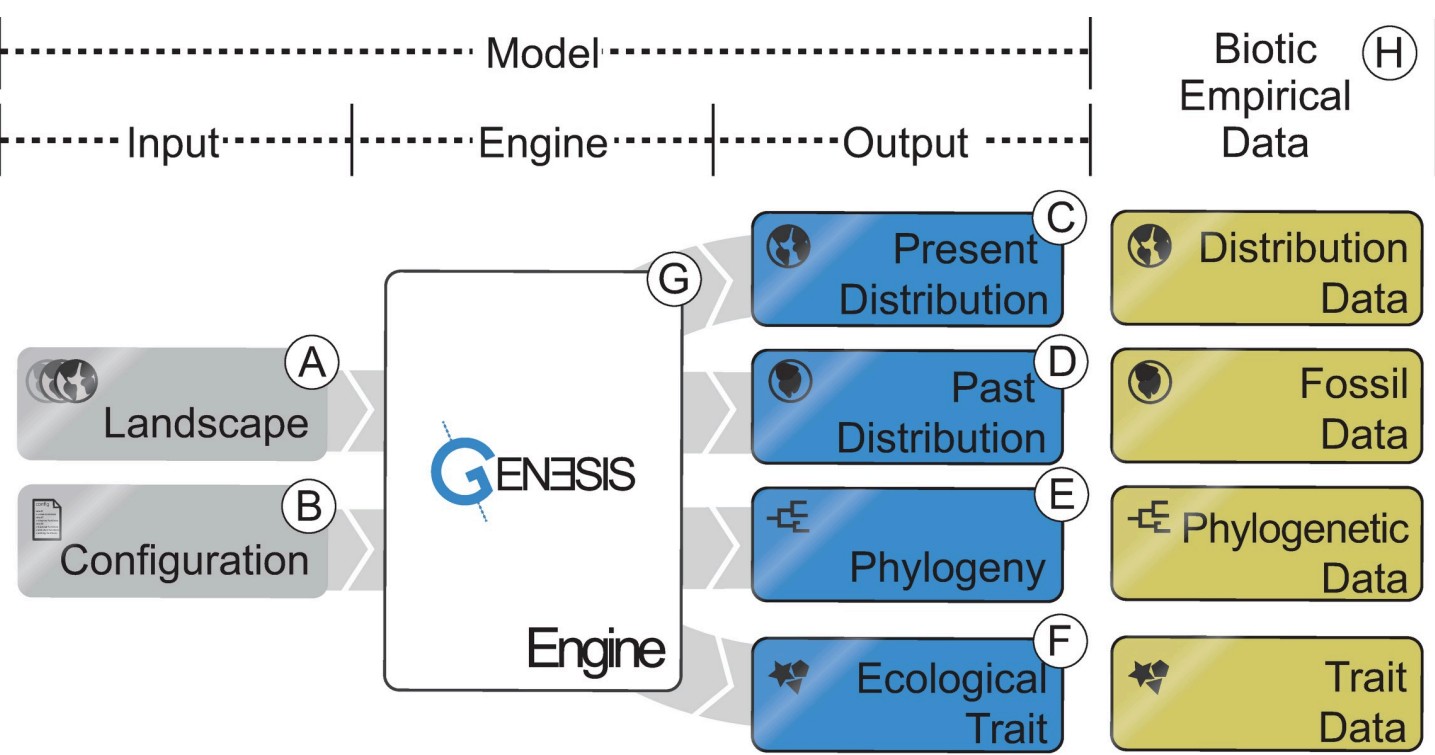

**Fig 1.** Schematic of the main components of the computer model: (A, B) model inputs, including the spatiotemporal landscape objects and the configuration file; (C–F) model outputs, including present and past species ranges, phylogenetic relationships among species, and the ecological traits of species; (G) model engine containing the mechanics; and (H) empirical data applicable for model validation.

been used for handling and analyzing spatial data [70]. Gen3sis follows best practices for scientific computing [71], including high modularization; consistent naming, style, and formatting; single and meaningful authoritative representation; automated workflows; version control; continuous integration; and extensive documentation.

Gen3sis operates over a grid-based landscape, either the entire globe or a specific region. The landscape used as input is defined by the shape of the colonizable habitat (e.g., land masses for terrestrial organisms), its environmental properties (e.g., temperature and aridity), and its connectivity to dispersal (e.g., the influence of barriers, such as rivers and oceans for terrestrial organisms). Gen3sis simulates species' population range dynamics, traits, diversification, and spatial biodiversity patterns in response to geological, biological, and environmental drivers. Using a combined trait-based and biological species concept, gen3sis tracks the creation, dynamics, and extinction of species ranges, which are composed of a set of sites occupied by species populations. Eco-evolutionary dynamics are driven by user-specified landscapes and processes, including ecology, dispersal, speciation, and evolution (Fig 1). Below, we explain the gen3sis inputs, the configurations (including eco-evolutionary processes), and the landscapes defining the computer model, as well as user-defined outputs (Fig 1C–1F).

## Inputs and initialization

Gen3sis has 2 input objects, which define a particular model (Fig 1). These inputs are (i) a dynamic landscape (Fig 1A), which is further divided into environmental variables and distance matrices; and (ii) a configuration (Fig 1B), in which the user can define initial conditions, biological functions, and their parameter values, as well as technical settings for the model core.

**Table 1. Presentation of the core functions of *speciation, dispersal, ecology,* and *evolution* implemented in gen3sis.** The computation of core functions is customizable in the configuration object. Shown are input objects that are combined to generate updated outputs. The table corresponds to the mechanisms presented in Fig 2B.

| Objective | Input | Computation | Output |
|---|---|---|---|
| **Speciation** | | | |
| Determines the divergence between geographic clusters of populations within a species; determines cladogenesis. | Species divergence matrix; species trait matrix; species abundance matrix; landscape values; distance matrix. | Divergence between geographically isolated clusters of populations increases over time, while (re) connected clusters decrease down to zero; speciation happens when the divergence between 2 clusters is above the speciation threshold, but it can also consider trait differences. | Updated species divergence matrix; new species if speciation occurred; updated genealogy table. |
| **Dispersal** | | | |
| Determines the colonization of vacant sites. | Species trait matrix; species abundance matrix; landscape values; distance matrix. | Species disperse according to a unique value or a distribution of dispersal values. | Updated species abundance matrix. |
| **Evolution** | | | |
| Determines the change in species traits in each site, anagenesis. | Species trait matrix; species abundance matrix; landscape values; geographic clusters; distance matrix. | Traits might change for each species in the populations of occupied sites. | Updated species trait matrix. |
| **Ecology** | | | |
| Determines the species abundance in each site. | Species trait matrix; species abundance matrix; landscape values; genealogy. | Change the species abundance, based on landscape environmental values and species co-occurrences; changes species trait values. | Updated species abundance matrix. |

## Landscape

The landscape objects (Fig 1A) form the spatiotemporal context in which the processes of speciation, dispersal, evolution, and ecology take place. Landscape objects are generated based on temporal sequences of landscapes in the form of raster files, which are summarized in 2 classes. The first landscape class contains (i) the geographic coordinates of the landscape sites; (ii) the corresponding information on which sites are generally suitable for a clade (e.g., land or ocean); and (iii) the environmental conditions (e.g., temperature and aridity). The landscape may be simplified into a single geographic axis (e.g., [72]) for theoretical experiments, or it may consider realistic configurations aimed at reproducing real local or global landscapes [26,73,74]. The second landscape class defines the connectivity of the landscape, used for computing dispersal and consequently isolation of populations. By default, the connection cost between occupied sites is computed for each time step from the gridded landscape data based on haversine geographic distances. This can be modified by a user-defined cost function in order to account for barriers with different strengths (e.g., based on elevation [73], water, or land) or even to facilitate dispersal in specific directions (e.g., to account for currents and river flow directions). The final connection costs are stored as sparse distance matrices [75]. Distance matrices, containing the connection costs, are provided at every time step as either (i) a precomputed full distance matrix, containing all habitable sites in the landscape (faster simulations but more storage required); or (ii) a local distance matrix, computed from neighboring site distances up to a user-defined range limit (slower simulation runs but less storage required).

## Configuration

The configuration object (Fig 1B) includes the customizable *initialization, observer, speciation, dispersal, evolution,* and *ecology* functions. These 6 functions define a configuration applied in the simulation engine (Table 1). The possibility to customize these functions confers the high flexibility of gen3sis by including a wide range of mechanisms, as illustrated by 5

configurations presented in a case study (Table A in S1 Note). Additionally, the configuration object lists the model settings, including (i) whether a random seed is used, allowing simulation reproducibility; (ii) start and end times of the simulation; (iii) rules about aborting the simulation, including the maximum global or local species number allowed; and (iv) the list of ecological traits considered in the simulation. One or multiple traits can be defined, which should correspond to those used in the *ecology* function. Moreover, the *initialization* function creates the ancestor species at the start of the simulation. Users can define the number of ancestor species, their distribution within the ancient landscape and their initial trait values. With the *observer* function, changes over time in any abiotic or biotic information of the virtual world can be recorded by defining the outputs that are saved at specified time steps. Outputs can be saved and plotted in real time as the model runs. The core biological functions (i.e. *speciation*, *dispersal*, *evolution*, and *ecology*) are presented below.

## Core functions and objects

The states of the model runs are updated in discrete time steps. At each time step, the *speciation*, *dispersal*, *evolution*, and *ecology* functions are executed sequentially (Fig 2). Speciation and extinction emerge from interactions across core functions. For example, speciation events are influenced by the *speciation* function, as well as by the *ecology* and *dispersal* functions that interact in a dynamic landscape, ultimately dictating populations' geographic isolation. Likewise, global extinctions depend on local extinctions, which decrease the number of inhabited sites until no sites remain inhabited by a species, rendering it extinct. Extinction happens when the occupied sites become uninhabitable and no other suitable sites are within dispersal distance or according to the change in species traits, rendering the species unfit for the environment. Internally, the computer model defines core objects of the simulations: species abundances; species trait values; the species divergence matrix between all populations for each species; and the phylogeny of all species created during the simulation. In the following sections, we describe the core processes in gen3sis, as well as their inputs and outputs. For a summary, see Table 1.

Running a simulation in gen3sis consists of the following steps: (i) Read in the configuration object, prepare the output directories, load the initial landscape (Fig 2A), and create the ancestor specie(s) (using the *initialization* function *create_ancestor_species*). (ii) Run the main loop over the landscape time steps. At every time step, the engine loads the appropriate landscape, removes all sites that became uninhabitable in the new time step, and executes the core functions as defined by the configuration object (Fig 2B). (iii) At the end of every time step, gen3sis saves the species richness, genealogy, and, if desired, the species, landscape, and other customized observations that are defined in the *observer* function (e.g., summary statistics and species pattern plots). Core functions are modifiable and can account for a wide range of mechanisms, as illustrated in the case study (S1 and S2 Notes). Conversely, functions can be turned off, for example, in an ecologically neutral model. For a pseudo-code of gen3sis, see S3 Note.

**Speciation. Core.** The *speciation* function iterates over every species separately, registers populations' geographic occupancy (species range), and determines when geographic isolation between population clusters is sufficient to trigger a lineage-splitting event of cladogenesis. A species' range can be segregated into spatially discontinuous geographic clusters of sites and is determined by multiple other processes. The clustering of occupied sites is based on the species' dispersal capacity and the landscape connection costs. Over time, disconnected clusters gradually accumulate incompatibility (divergence), analogous to genetic differentiation. Disconnected species population clusters that maintain geographic isolation for a prolonged

## A Landscape

## B Core processes

**Fig 2. Schematic example of the gen3sis engine simulation cycle of one species' populations over a landscape evolution example containing highlands (yellow), lowlands (green), and a river acting as a barrier (blue).** (A) Landscape. A time series of landscapes is used as input, with the landscape being updated after every time step of the simulation cycle, i.e., after the ecology process. (B) Model core processes. *First*, the speciation process determines the divergence between geographic clusters of populations that are not connected and splits the clusters into new species if a threshold is reached. In this illustration, divergence between clusters of the species' populations was not sufficient to trigger speciation. *Second*, in the dispersal process, the species spreads within a landscape to reachable new sites. In this illustration, the river limits dispersal. *Third*, the evolution process can modify the value of the traits in the populations. In this illustration, 2 populations show trait evolution in their ability to cope with the local environment (i.e., red and white populations). *Fourth*, the ecology process recalculates the abundance of the species in each site based on the abiotic condition and co-occurring species, possibly resulting in local extinctions. In this illustration, the red population was unsuited to the lowlands, while the white population survived in the highlands. Speciation and extinction events emerge from multiple simulation cycles of customizable processes.

period of time will result in different species after the differentiation threshold ϙ is reached (modeling Dobzhansky–Muller incompatibilities [76]). These clusters become 2 or more distinct species, and a divergence matrix reset follows. On the other hand, if geographic clusters come into secondary contact before the speciation occurs, they coalesce and incompatibilities are gradually reduced to zero.

**Nonexhaustive modification possibilities.** A customizable *speciation* function can further embrace processes that modulate speciation. Increased divergence values per time step can be constant for all species or change depending on biotic and abiotic conditions, such as faster divergence between species occupying higher temperature sites [64], or they can be dependent on population size [77] or other attributes [78]. The function also takes the ecological traits as input, thus allowing for ecological speciation [24], where speciation depends on the divergence of ecological traits between—but not within—clusters [79].

**Dispersal.   Core.** The *dispersal* function iterates over all species populations and determines the connectivity between sites and the colonization of new sites in the landscape. Dispersal distances are drawn following a user-defined dispersal function and then compared with the distance between pairs of occupied and unoccupied sites accounting for landscape costs. A unique dispersal value can be used (deterministic connection of sites) or dispersal values can be selected from a specified distribution (stochastic connection of sites). If the dispersal cost between the sites is lower than the dispersal value, the dispersal is successful. If populations from multiple sites of origin manage to reach an unoccupied site, the final colonizer is selected randomly to seed the newly occupied site.

**Nonexhaustive modification possibilities.** A customizable *dispersal* function enables the modeling of different dispersal kernels depending on the type of organism considered. Dispersal values can be further linked with the *ecology* function, e.g., a trade-off with other traits [80] and dispersal versus competitive ability [81], and the *evolution* function allowing dispersal to evolve, resulting in species with different dispersal abilities [82].

**Evolution.   Core.** The *evolution* function determines the change in the traits of each population in occupied sites of each species. Traits are defined in the configuration object and can evolve over time for each species' populations. The function iterates over every population of a species and modifies the trait(s) according to the specified function (e.g., traits related to dispersal, niche, or competition).

**Nonexhaustive modification possibilities.** A customizable *evolution* function takes as input the species abundance, species trait, species divergence clusters, and landscape values. In the function, it is possible to define which traits evolve and how they change at each time step. In particular, the frequency and/or amount of change can be made dependent on temperature [83], ecological traits [84], or abundances [85], while the directions of change can follow local optima or various evolutionary models, including Brownian motion [86] and Ornstein–Uhlenbeck [87].

**Ecology.   Core.** The *ecology* function determines the abundance or presence of populations in occupied sites of each species. Thus, extinction processes derive from *ecology* function interactions with other processes and landscape dynamics. The function iterates over all occupied sites and updates the species population abundances or presences on the basis of local environmental values, updated co-occurrence patterns, and species traits.

**Nonexhaustive modification possibilities.** A customizable *ecology* function takes as input the species abundance, species trait, species divergence and clusters, and the landscape values. Inspired by classic niche theory [10,15,88], the function can account for various niche mechanisms, from simple environmental limits to complex multispecies interactions. It is possible, for example, to include a carrying capacity for the total number of individuals or species [21] or competition between species based on phylogenetic or trait distances [23], based on an interaction currency [89], or further constrained by a functional trade-off [80].

## Outputs and comparisons with empirical data

The computer model delivers a wide range of outputs that can be compared with empirical data (Fig 1, Table 2). Gen3sis is therefore suitable for analyzing the links between interacting processes and their multidimensional emergent patterns. By recording the time and origin of all speciation events, as well as trait distributions and abundance throughout evolutionary history, the simulation model records the information required to track the dynamics of species diversity and the shaping of phylogenetic trees. The most common patterns observed and studied by ecologists and evolutionary biologists, including species ranges, abundances, richness, and genealogies, are emergent properties of the modeled processes (Table 2). All internal

**Table 2. List of outputs from the gen3sis computer model, both direct and indirect, that can be compared with empirical data.** Direct outputs are the species abundance matrix, species trait matrix, and phylogeny, while indirect outputs result from various combinations of the direct outputs. The computations of indirect outputs rely on other packages available in the R environment [67].

| Pattern | | Scale | | | | | |
| --- | --- | --- | --- | --- | --- | --- | --- |
| | | Spatial | | | Temporal | | |
| | | − | I | + | − | I | + |
| Metric | Example | local | regional | global | present | past | deep past |
| Alpha diversity (α) | Local species richness follows marked spatial gradients, such as along latitude (LDG [91]). Species richness is further correlated across scales when the regional species pool size is positively associated with local species richness (e.g., [4,92]). | * | * | * | * | * | * |
| Beta diversity (β) | Species turnover is marked along both spatial and environmental gradients [93,94] and can display sharp boundaries forming biogeographic domains [95]. | | * | * | * | * | * |
| Gamma diversity (γ) | Regional difference in species richness, e.g., across biogeographic regions with comparable climates, such as the continental temperate region of North America versus Asia [96]. | | | * | * | * | * |
| Species abundance, frequency, and range | Assemblages are generally composed of a few very abundant species and many rare species [97,98]. A few species tend to occupy many sites, while most are very rare and have a restricted range size [99]. | * | * | * | * | * | * |
| Species ecological niche width distribution | Niche width is heterogeneous across species [100,101], and narrow niche width leads to higher speciation [102]. | * | * | * | * | * | * |
| Trait evolutionary rates | Ecological traits and niches generally evolve slowly so that closely related lineages have similar traits and niches, coined as niche conservatism [60]. | * | * | * | * | * | * |
| Species diversification rates | Species diversification rate varies over time and across clades [103–105]. | * | * | * | | * | * |
| Topological and temporal phylogenetic properties | Empirical phylogenetic trees typically display a topological signature [106] and have more divided branching over time, with marked prevalence of a recent branching distribution [107]. | * | * | * | | * | * |
| Phylogenetic alpha (α) and beta (β) diversity | Local communities can show either phylogenetic overdispersion or clustering compared with the regional pool [108]; greater geographic distances correspond to increased phylogenetic β diversity across biogeographic barriers [109]; decay in phylogenetic similarity with increasing geographic distance [110]. | * | * | * | | * | * |
| Functional alpha (α) and beta (β) diversity | Local assemblages represent a subset of the regional functional diversity; functional traits show a typical turnover spatially, often along environmental gradients [111]. | * | * | * | * | * | * |

LDG, latitudinal diversity gradient.

objects are accessible to the observer function, which is configurable and executed during simulation runs. Gen3sis provides direct simulation outputs in a format ready to be stored, analyzed, and compared with empirical data. Given the flexibility of gen3sis, it is possible to explore not only parameter ranges guided by prior knowledge available for a given taxonomic group, but also a variety of landscape scenarios and mechanisms (Fig 3). Furthermore, to attain generality, validating modeled outputs with multiple empirical patterns is recommended [20,25,35]. Gen3sis generates multiple outputs, which can be compared with empirical data using simulation rankings or acceptance criteria [25,35,90].

## Case study: The emergence of the LDG in the Cenozoic

### Context

Expanding from previous studies using mechanistic simulation models [22,23,39], we use a case study to illustrate the flexibility of gen3sis to implement mechanisms that derive from a variety of biological hypotheses, as well as null models to serve as contrast. We present how gen3sis can be used to explore a variety of proposed mechanisms and how to explore parameters, but the case study is not a comprehensive exploration of existing hypotheses and their associated parameters [20].

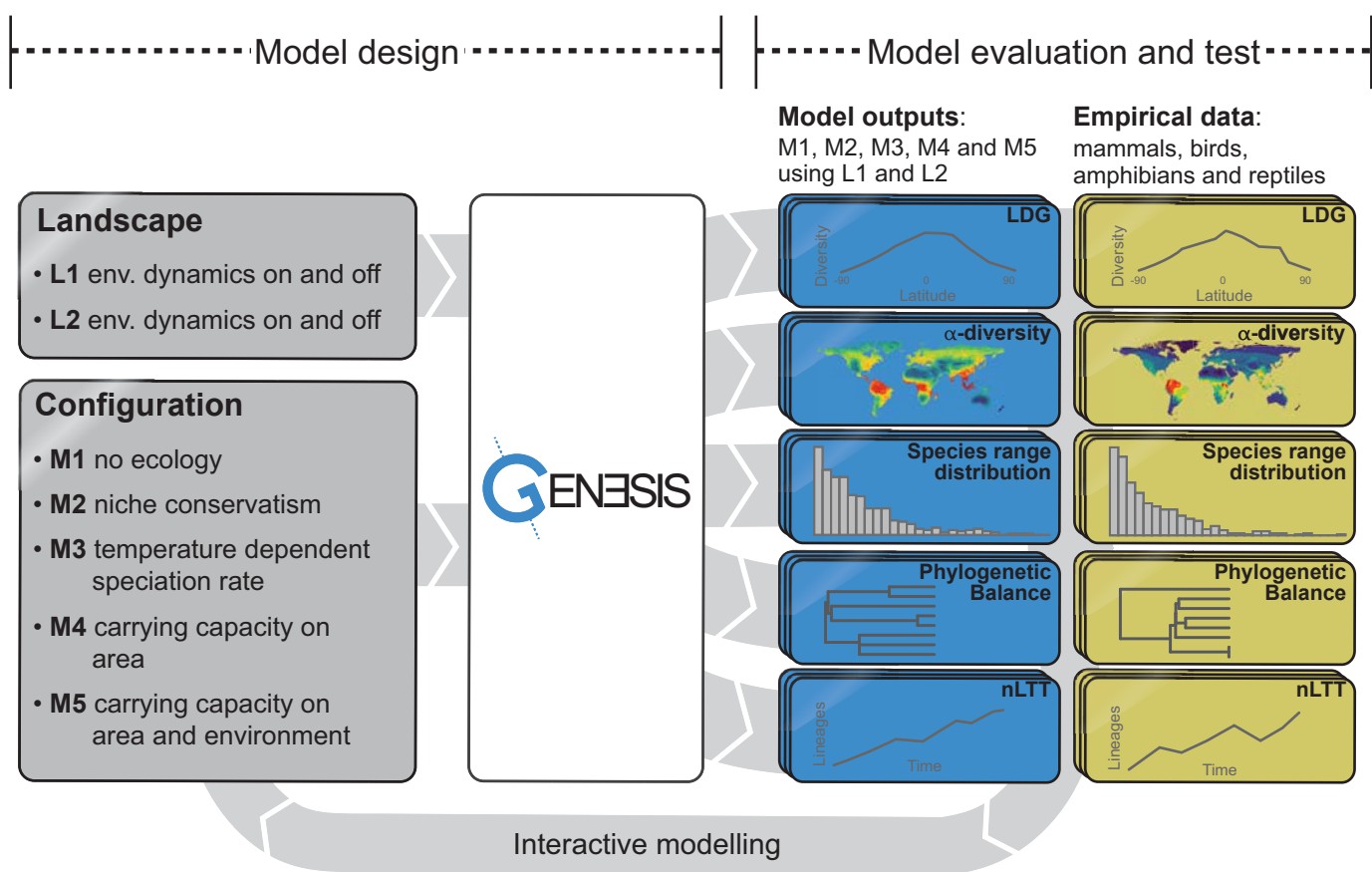

**Fig 3. Schematic representation of the case study showing the model design with 2 landscapes (i.e., L1 and L2, with and without temporal environmental variability) and configurations of 5 models (i.e., M1, M2, M3, M4, and M5; Table A in S1 Note) and model evaluation and testing, based on multiple patterns including LDG, spatial α-diversity, range size distributions, phylogenetic balance, and temporal dynamics of species diversification (nLTT).** Selection criteria were based on empirical data from major tetrapod groups, i.e. mammals, birds, amphibians and reptiles (Table 3). LDG, latitudinal diversity gradient; nLTT, normalized lineage though time.

The LDG is one of Earth's most iconic biodiversity patterns, but the underlying mechanisms remain largely debated [20,63,64,101,102,112–114]. Many hypotheses have been proposed to explain the formation of the LDG [20], and these generally agree that a combination of biological processes and landscape dynamics has shaped the emergence of the LDG [20]. Among the proposed hypotheses, it has been postulated that older and more stable tropical environments have more time for cumulating species and have reduced extinctions, while niche conservatism limits the spread of lineages to more recent colder environments [59–62]. Second, higher temperatures in the tropics increase metabolic and mutation rates, which could lead to faster reproductive incompatibilities among populations and higher speciation rates compared with colder environments [63,64]. Third, the tropics are generally more productive than colder environments and greater resource availability can sustain higher abundances, and, therefore, a larger number of species can coexist there [65,66,115,116]. From these hypotheses, we illustrate 5 derivative models in gen3sis: a null model without ecological filtering or trait evolution (M1); a model of trait evolution only considering niche conservatism where trait evolutionary rates are limited (M2); a model where evolutionary rates are proportional to the occupied site temperatures (M3); a model with uniform carrying capacity (M4); and a model where carrying capacity depends on temperature and aridity (M5). We

simulated the spread, speciation, dispersal, and extinction of terrestrial organisms over the Cenozoic marked by various shifts in diversification. Finally, we evaluated whether the emerging patterns from these simulated mechanisms correspond to the empirical LDG, phylogenetic tree imbalance, and range size frequencies computed from data of major tetrapod groups, including mammals, birds, amphibians, and reptiles (Fig 3).

### Input landscapes

The quality of the outputs of simulation models such as gen3sis hinges on accurate and relevant reconstructions of past environmental conditions [117]. Conditions during the Cenozoic (i.e., 65 Ma until the present) are considered key for the diversification of the current biota [118], and the Cenozoic is the period during which the modern LDG is expected to have been formed [119]. In the Cenozoic, the continents assumed their modern geographic configuration [26]. Climatically, this period was characterized by a general cooling, especially in the Miocene, and ended with the climatic oscillations of the Quaternary [120].

We compiled 2 global paleoenvironmental landscapes (i.e., L1 and L2) for the Cenozoic at 1° and approximately 170 kyr of spatial and temporal resolution, respectively (S1 Note, S1 and S2 Animations). To account for uncertainties in paleoreconstructions on the emerging large-scale biodiversity patterns, we used 2 paleoelevation reconstructions [121,122] associated with 2 approaches for estimating the paleotemperature of sites (S1 Note). L1 had temperatures defined by Köppen bands based on the geographic distribution of lithologic indicators of climate [56]. L2 had temperature defined by a composite of benthic foraminifer isotope records over time [123] and along latitude for specific time periods [124–130]. An aridity index ranging from 0 to 1 was computed based on the subtropical arid Köppen zone for both landscapes [56]. Finally, in order to test for the effects of deep-time environmental dynamics, we also ran simulations (i.e., L1.0 and L2.0) in which a constant contemporary landscape was set for the same number of time steps as in L1 and L2 (S1 Note).

We used available paleoelevation models [121,122] and paleoclimate indicators [56,123–133] to generate input landscapes to explore the formation of the LDG and account for uncertainties and limitations. Hence, the case study represents an illustration of how gen3sis can handle multiple reconstructions that interact with eco-evolutionary processes in complex ways. Further research in geology and climatology is required to generate more accurate paleolandscapes than those presented here.

### Model configurations

We implemented 5 illustrative gen3sis models derived from hypotheses on the emergence of the LDG. The models (i.e., M1, M2, M3, M4, and M5) had distinct speciation and ecological processes and contrast the common idea that time, diversification rates, and ecological limits underpin the LDG (Fig 3, Table A in S1 Note). As a simplified approach for this illustration, all simulations were initiated with one single ancestor species spread over the entire terrestrial surface of the Earth at 65 Ma [134], but initial conditions could also match the ancestral range informed by fossil records [47]. The temperature optimum of each population was initiated to match local site conditions. Since we focused on terrestrial organisms, aquatic sites were considered inhabitable and twice as difficult to cross as terrestrial sites. This approximates the different dispersal limitations imposed by aquatic and terrestrial sites. To compute the full distance matrix, we used haversine geodesic distances.

**M1.** We applied a null model where all the terrestrial sites were ecologically equivalent. Temperature and aridity thus did not determine the niche of the species. The divergence rate between isolated clusters was kept constant (i.e., +1 for every 170 kyr of isolation). Clusters of

populations that accumulated differentiation over time speciated according to a speciation threshold φ. Ecology and trait evolution were turned off, making this null model a baseline with which all the other more complex models can be contrasted.

**M2.** In the implementation of the *niche conservatism*, the *ecology* function defined the species population abundance, where the abundance increased proportionally to the distance between the population temperature niche optimum and the site temperature (S1 Note). The temperature optimum of each population was set to evolve randomly, with a normal distribution following Brownian motion with standard deviation $\sigma^2$.

**M3.** In the implementation of the *diversification rates*, the speciation function applied a temperature-dependent divergence between population clusters [63,64]. Species in warmer environments accumulated divergence between disconnected clusters of populations at a higher rate (S1 Note). The accumulation of divergence was set to be 3 times faster at the warmest sites. The rate of differentiation increase was shaped by the average site temperature of the species clusters to the power of $d_{power}$ plus a constant. Overall, this created higher speciation rates at warmer than at colder sites (S1 Note, S1 Fig).

**M4.** In the implementation of the *carrying capacity*, we applied a model where the total number of individuals was equally limited in each site, as an overall constraint on biotic interactions [135]. Because of resource and space limitations, only a limited number of individuals ($k$) could coexist within the site. If the sum of all species abundances in a site was above $k$ (modulated by $k_{power}$), species abundances were randomly reduced across species until $k$ was reached. This contrasts with the next model M5, where the carrying capacity varied with temperature and aridity. Locally extinct species were the ones with zero individuals after the limit $k$ was applied (S1 Note).

**M5.** In the implementation of the *ecological limits*, the *ecology* function included a carrying capacity $k$ of each site that scaled with area energy (i.e., temperature and aridity) [116,136]. In this model, we assumed that the carrying capacity of the number of individuals at sites scaled with energy, which indirectly also constrained the number of species that could coexist in a given place [21,116]. If the sum of all species abundances in a site was above $k$ (modulated by $k_{power}$), species abundances were randomly reduced across species until $k$ was reached, as in M4 (S1 Note).

## Exploration of model parameters

We explored the parameter space of each model using Sobol sequences, a quasi-random number generator that samples parameters evenly across the parameter space [137]. We explored parameter ranges by basing upper and lower parameter boundaries on the literature and interactive modeling explorations. In all models, species dispersed following a Weibull distribution with shape $\phi$ = [2 to 5] and a scale of $\Psi$ = [550 to 850], resulting in most values being around 500 to 1,500 km, with rare large dispersal events above 2,000 km. The explored dispersal distribution parameters ranged in realized mean and 95% quantiles between less than a single cell, i.e., approximately 50 km for a landscape at 4°, and more than the Earth's diameter, i.e., approximately 12,742 km (S2 Fig). In all models except M1, the *evolution* function defined the temperature niche optimum to evolve following Brownian motion. The temperature optimum of each population was set to evolve randomly, following a normal distribution in a Brownian motion fashion with standard deviation $\sigma^2$ = [0.001 to 0.010], corresponding to [±0.1°C to ±1°C] per time step. In all models except M3, species emerged after φ = [6 to 60], corresponding to events occurring after [1 to 10] myr of isolation in the cases where the divergence rate was kept constant. For M3, the differentiation increase with temperature (i.e., 3 times faster at the hottest sites) changed to the power of $d_{power}$ = [2 to 6] plus a constant (S1 Fig).

Temperature niche optima were homogenized per geographic cluster by an abundance-weighted mean after ecological processes happened. Carrying capacities had $k$ values ranging from low to high with a power-law scaling $k_{power}$ = [1 to 4]. For further details on the simulation model framework, model parameters, initial conditions, paleoenvironmental reconstructions, and landscape modification experiment, see S1 Note and Table A in S1 Note.

For each model (i.e., M1, M2, M3, M4, and M5) in combination with each landscape (i.e., L1 and L2) with and without deep-time environmental dynamics, we ran a full factorial exploration of these parameter ranges at a coarse resolution of 4˚ (i.e., M1 $n$ = 300, M2 $n$ = 780, M3 $n$ = 1,020, M4 $n$ = 300, M5 $n$ = 780) and compared these to empirical data. Simulations considered further (i) had at least 20 species at the present; (ii) had fewer than 50,000 species; or (iii) had fewer than 10,000 species cohabiting the same site at any point in time (S1 Note). After parameter range exploration, we identified realistic parameters and ran a subset at 1˚ for high-resolution outputs for illustration (Fig 4). Parameter exploration is illustrative and could be expanded in future research applications.

## Correspondence with empirical data

We compared simulation ability to produce the observed biodiversity patterns using a pattern-oriented modeling (POM) approach [25,90]. POM compares the predictions of each model and parameter combination with a number of diagnostic patterns from empirical observations. In our case, we used the LDG slope and curve, spatial α-biodiversity, range size frequencies, tree imbalance, and macroevolutionary temporal dynamics as diagnostic patterns (Fig 3, S1 Note). The POM approach allows a calibration and model comparison based on high-level diagnostic patterns, avoiding the hurdles of defining explicit (approximate) likelihood functions [138]. The POM approach requires the specification of a range for each pattern under which observation and prediction are accepted, hence when a simulation satisfactorily reproduces empirical observations. Unless POM is coupled with an explicit probabilistic model [138], the limits for acceptance must be decided based on the empirical data distribution [25,90]. In complement to POM, we computed the Bayesian information criteria (BIC), balancing the fit of the model to α-diversity with a penalization for model complexity (S1 Note).

To generate the empirical values for these patterns, we obtained distribution data on 25,941 species [139–141], following [142], and phylogenetic data on 18,978 species [5,143–146], following [147] for major tetrapod groups, i.e., terrestrial mammals, birds, amphibians, and reptiles (S1 Note). LDG$_{\%loss}$ was defined as the percentage of species loss per latitudinal degree and was measured as the slope of a linear regression of normalized species richness against absolute latitude. The β-statistics [31] was used for phylogenetic tree imbalance in ultrametric trees, following [106]. Species range decrease (SRD) in km$^2$ was defined as the percentage of species loss per species range and was measured as the slope of a linear regression of range size distributions. We further compared the mean species number per latitude curve (LDG$_{curve}$), normalized lineage though time (nLTT) curves, and α-biodiversity spatial distribution (S1 Note). Empirical values of LDG, β, and SRD were as follows: mammals (LDG = 5.1%, β = −0.4, SRD = 2.3*10$^3$%), birds (LDG = 1.5%, β = −1.3, SRD = 6.5*10$^7$%), amphibians (LDG = 3.9%, β = −0.7, SRD = 0.11%), and reptiles (LDG = 1.5%, β = −0.8, SRD = 5.3*10$^3$%). Based on these values, we used the following acceptance criteria: (i) LDG between 5.4% and 1.1%; (ii) tree shape statistic β between −1.4 and −0.3; (iii) range size frequencies with a decrease in the number of large-range species with a tolerance of 5% [97–99]; (iv) correlation of mean species number per latitude with r > 0.4; and (iv) nLTT curve difference < 0.15.

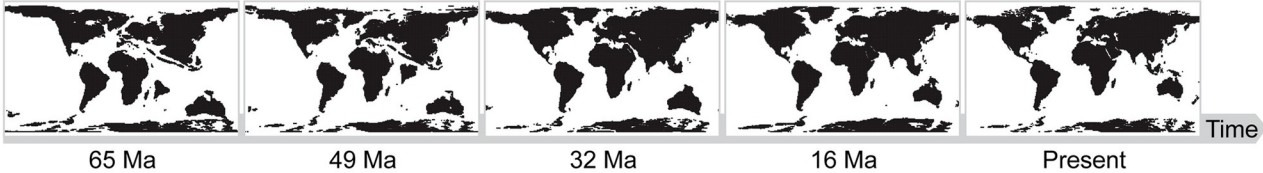

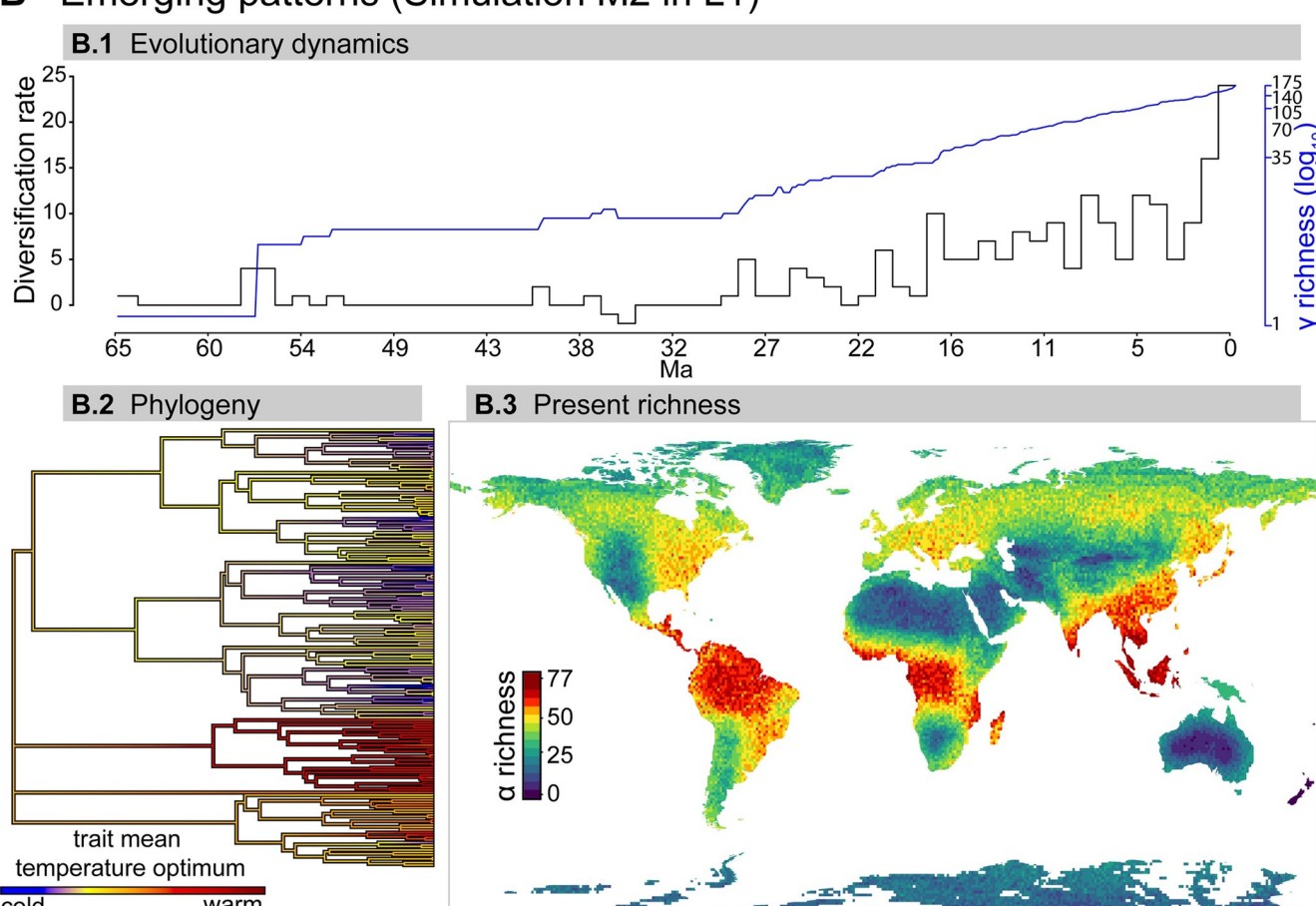

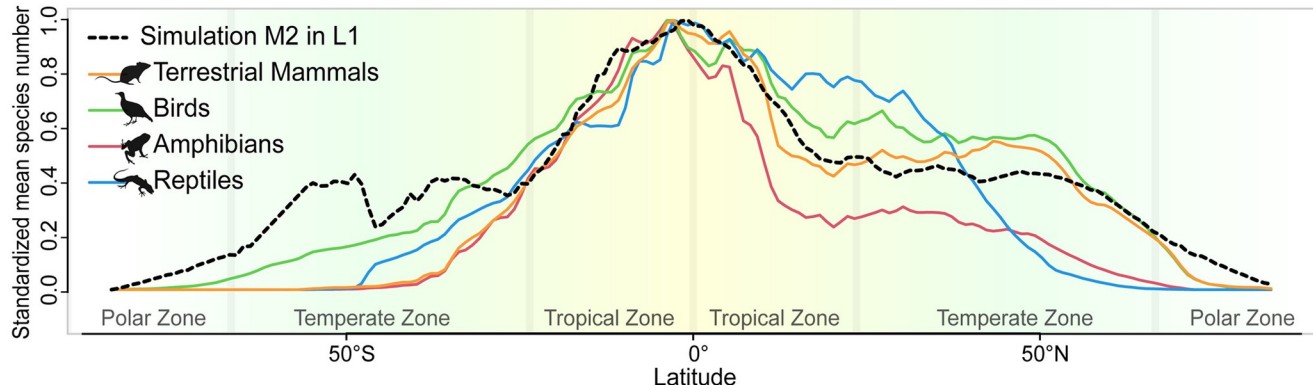

**Fig 4. Illustration of one global simulation of the speciation, dispersal, and extinction of lineages over the Cenozoic, starting with a single ancestor species and imposed energetic carrying capacity (M5 in L1).** We selected the best matching simulation of M5 in L1 at 1° (**n = 12**) that predicted realistic biodiversity patterns. (A) Images of the Earth land masses through time, used as input for the simulation. (B) Selected emerging patterns: evolutionary dynamics, phylogeny, and present richness. (B.1) Evolutionary dynamics: γ richness (log$_{10}$ scale) through time (blue line) and diversification rate. (B.2) Phylogeny showing the distribution of the temperature optima for all extant species. (B.3) Present distribution of simulated α biodiversity globally, which indicates locations of biodiversity hotspots. For the empirical match, see S8 Fig. (C) Model correspondence with empirical data on terrestrial mammals, birds, amphibians, and reptiles for the LDG, measured as the standardized and area-scaled mean species number per latitudinal degree. The emerging LDG$_{\%loss}$ (i.e., 4.6% of species loss per latitudinal degree) closely matched empirical curves, with good agreement for mammals (Pearson r = 0.6), birds (r = 0.57), amphibians (r = 0.57), and reptiles (r = 0.38) (S1 Note, Figs 4C and S9). Data presented here are available in S1 Data at https://zenodo.org/record/5006413, including selected simulation summary output (phylogeny and richness) and empirical richness used to derive LDG curves. LDG, latitudinal diversity gradient.

## Simulations results

We illustrate how integrating deep-time environmental dynamics and biological processes can help us understand the origin of biodiversity patterns. Simulations including deep-time temporal dynamics systematically showed a best fit to empirical data (S4 Fig). Models M4 and M5 resulted in the best match for most of the empirical patterns individually, and M5 was the only model able to pass all acceptance criteria (Table 3). Although all 5 models were able to reproduce the LDG$_{\%loss}$, M5 was superior in explaining the LDG$_{curve}$ (S5 and S6 Figs), α-diversity, phylogenetic tree imbalance, and species range size frequencies simultaneously (Table 3). Most simulations of model M5 (67%) resulted in a decrease in species richness at higher latitudes, indicating that the LDG emerged systematically under M5 mechanisms (S8 Fig, Tables B, C, and D in S1 Note). Using the BIC approach, and accounting for model complexity, we found that the models implementing carrying capacities (i.e., M4 and M5) were the only ones significantly superior to the null model M1 when considering α-diversity spatial patterns (S3, S5, and S6 Figs). Finally, we found that the support for M4 and M5 over M1, M2, and M3 was consistent across the 2 alternative landscapes L1 and L2 (S6 and S8 Figs, Table D in S1 Note). We further illustrate the capacity to run high-resolution simulations for a subset of the

**Table 3. Model acceptance table with pattern descriptions and details of acceptance derived from empirical data.** Percentages of accepted simulations (for both landscapes) are shown for each model and acceptance parameter and the combination of all acceptance patterns. For details, see S1 Note.

| Acceptance | | M1 | M2 | M3 | M4 | M5 |
|---|---|---|---|---|---|---|
| Pattern | Description and empirical acceptance | **n = 300** | **n = 780** | **n = 1,020** | **n = 300** | **n = 780** |
| **LDG$_{\%loss}$** | Percentage of species loss per latitudinal degree from linear regression slope.<br>Accept LDGs between 5% and 1% | 26% | 28% | 41% | 34% | 36% |
| **LDG$_{curve}$** | Standardized mean species number per latitude correlation between simulated and empirical maximal Pearson correlation.<br>Accept r > 0.4 | 21% | 28% | 43% | 38% | 60% |
| **α biodiversity** | Spatial distribution correlation between simulated and empirical maximal Pearson correlation.<br>Accept r > 0.4 | 18% | 24% | 37% | 20% | 36% |
| **Range** | Range size distributions.<br>Accept only distributions that show a consistent frequency decrease toward large-ranged species with a tolerance of 5% | 23% | 8% | 4% | 31% | 16% |
| **Phytogenic balance** | The imbalance of a phylogenetic tree is measured by the value that maximizes the likelihood in the β-splitting model [152].<br>Accept phylogenies with β between −1.4 and −0.3 | 56% | 56% | 55% | 73% | 64% |
| **nLTT** | Temporal dynamics of species diversification, measured by the differences between empirical and simulated nLTT curves [153].<br>Accept nLTT differences < 0.15 | 66% | 65% | 70% | 62% | 59% |
| **Combined** | Simulations passing all criteria above with at least 20 species alive at present time | 0% | 0% | 0% | 0% | 1% |

LDG, latitudinal diversity gradient; nLTT, normalized lineage though time.

explored parameters. Increasing the spatial resolution of the simulations ($n = 12$) resulted in an increase in γ richness and computation time and a slight decrease in the LDG$_{\%loss}$ (S7 Fig), which was associated with a disproportionally larger number of sites toward higher latitudes, which, in turn, affected population connectivity and, therefore, speciation rates [148].

In order to gain insight into the eco-evolutionary processes leading to the simulated patterns, we quantified the speciation, extinction, and migration rates within and between low (23° 27′ N- 23° 27′ S) and high (66° 33′ N -23° 27′ N; 23° 27′ S-66° 33′ S] latitudes from the best ranking simulations. Speciation and extinction rates were consistently higher at low compared with high latitude (S10 Fig), but speciation was systematically superior to extinction in contributing to the LDG. In contrast, dispersal from low to high latitude was always more frequent than from high to low latitude (Table F in S1 Note), which contributed to attenuation of the LDG. Because diversity was higher in the tropics, species were more likely to move from low to high latitude, corroborating empirical observations [149]. Moreover, our results indicate that an increase in the scaling factor of carrying capacity with energy k resulted in a steeper LDG$_{\%loss}$ (Tables B and C in S1 Note), which is in agreement with findings from previous studies [21,63,116,136]. Similarly, increasing the time for divergence consistently led to lower species richness and flattened the LDG slope so that the tropics accumulated diversity more slowly, but changes in speciation rates were less likely to drive large-scale biodiversity patterns [114]. Including a carrying capacity led to a characteristic increase in speciation and extinction rates toward the present, which intensified when temperature and aridity were considered as limiting factors (S10–S12 Figs), matching the recent diversification found in empirical data [150].

## Synthesis

In accordance with Rangel and colleagues [23], we found that realistic LDG patterns are dependent on species evolutionary responses to environmental dynamics. Rangel and colleagues [23] concluded that LDG patterns in South America are sensitive to the rate of evolutionary adaptation to climatic factors, which are dynamic in time (climate oscillations) and space (topography). However, while in that study [23], the intensity of competition was assumed to be an inverse function of phylogenetic distance; in gen3sis, competition can be modeled directly through traits and carrying capacity, opening up a new pathway for future investigations. In addition, Saupe and colleagues [22] showed that simulations with poor dispersal are better at representing the observed strong LDG in tetrapods. In agreement with their results, our parameter explorations indicated that dispersal correlated negatively with LDG [22], and simulations with lower dispersal parameters agreed better with the data (S1 Note). While previous case studies using computer models have conveyed information on the formation of the LDG [22,23,39], they used a shorter timeframe (i.e., below 1 Ma) and/or explored few mechanisms, i.e., a simplified landscape or a single acceptance criterion [26,38,43,114]. Although our case study was illustrative and we implemented only a small representative fraction of the candidate processes and parameters expected to shape biodiversity patterns, it illustrates how gen3sis can handle multiple interacting eco-evolutionary processes proposed in the literature. Still, it is imperative that future research explore further mechanisms and parameters combinations in order to advance our understanding of the processes behind the emergence of biodiversity.

Although recent studies using realistic landscapes and computer models reproduced biodiversity patterns over a time scale spanning the Quaternary [22,23,39], many speciation and extinction events shaping present diversity patterns date back before the glaciation, and few studies have covered deep-time dynamics [26,38,43,142]. Deep-time landscape reconstructions

are still generally lacking but are increasingly becoming available [121,123]. For example, we represented Quaternary climatic oscillation using approximately 170 kyr time steps, which correspond to a coarser temporal scale compared with the frequency of oscillations, and thus do not account for shorter climatic variation effects on diversity patterns [22,23,39]. We also did not consider ice cover, which can mask species' habitable sites, which might explain mismatches between simulated and empirical LDG patterns below 50˚ (Fig 4C). Moreover, paleoindicators of climate from Köppen bands have major limitations, and the temperature estimation derived in our case study might suffer from large inaccuracies. Lastly, extrapolation of the current temperature lapse rate along elevation might lead to erroneous estimates, especially in terms of the interaction with air moisture [151], which was not further investigated here. Through interdisciplinary research across the fields of geology, climatology, and biology, we expect that gen3sis will improve our understanding of the shaping of biodiversity across space, time, and complexity.

## Discussion

Understanding the emergence of biodiversity patterns requires the consideration of multiple biological processes and abiotic forces that potentially underpin them [20,23,35,36]. We have introduced gen3sis, a modular, spatially explicit, eco-evolutionary simulation engine implemented as an R package, which offers the possibility to explore ecological and macroevolutionary dynamics over changing landscapes. Gen3sis generates commonly observed diversity patterns and, thanks to its flexibility, enables the testing of a broad range of hypotheses (Table 4). It follows the principle of computer models from other fields [154–156], where mechanisms are implemented in a controlled numeric environment and emerging patterns can be compared with empirical data [25]. The combination of exploring patterns emerging from models and qualitatively and quantitatively matching their outputs to empirical data should increase our understanding of the processes underlying global biodiversity patterns.

**Table 4. A nonexhaustive list of expected applications of gen3sis.** Given the flexibility and the range of outputs produced by the engine, we expect that gen3sis will serve a large range of purposes, from testing a variety of theories and hypotheses to evaluating phylogenetic diversification methods.

| Use | Examples from Fig 1 |
|---|---|
| Testing phylogenetic inference methods, including diversification rates in phylogeographic reconstructions. | Infer diversification rate in gen3sis simulated phylogenies (E) and compare with a known diversification in gen3sis (A, B, and G). |
| Providing biotic scenarios for past responses to geodynamics. | Based on model outputs (C–F) and comparisons with empirical data (H), select plausible models (B). |
| Testing paleoclimatic and paleotopographic reconstructions using biodiversity data. | Based on model outputs (C–F) and comparisons with empirical data (H), select plausible landscape(s) (A). |
| Comparing expectations of different processes relating to the origin of biodiversity; generating and testing hypotheses. | Compare models (A, B, and G) with outputs (C–F) and possibly how well outputs match empirical data (H). |
| Comparing simulated intraspecific population structure with empirical genetic data. | Compare simulated divergence matrices with population genetic data. |
| Forecasting the response of biodiversity to global changes (e.g., climate or fragmentation). | Extrapolate plausible and validated models (A, B, and G) to landscapes under climate change scenarios (A). |
| Investigating trait evolution through space and time. | Combine past and present simulated species traits (F) and distributions (C, D) with fossil and trait data (H). |
| Modeling complex systems in space and time in unconventional biological contexts in order to investigate eco-evolutionary processes in fields traditionally not relying on biological principles. | Model eco-evolutionary mechanisms (A, B, and G) in an unconventional eco-evolutionary context. |

Verbal explanations of the main principles underlying the emergence of biodiversity are frequently proposed but are rarely quantified or readily generalized across study systems [20]. We anticipate that gen3sis will be particularly useful for exploring the consequences of mechanisms that so far have mostly been verbally defined. For example, the origins of biodiversity gradients have been associated with a variety of mechanisms [7], but these represent verbal abstractions of biological processes that are difficult to evaluate [20]. Whereas simulation models can always be improved, their formulation implies formalizing process-based abstractions via mechanisms expected to shape the emergent properties of a system [157]. Specifically, when conveying models with gen3sis, decisions regarding the biological processes and landscapes must be formalized in a reproducible fashion. By introducing gen3sis, we encourage a standardization of configuration and landscape objects, which will facilitate future model comparisons. This standardization offers a robust framework for developing, testing, comparing, and applying the mechanisms relevant to biodiversity research. Moreover, modeling eco-evolutionary processes in a flexible platform enables the exploration of how biodiversity statistics may depend on a multitude of different model assumptions and parameter values. This approximates how biodiversity patterns relate to eco-evolutionary processes. Further studies exploring the dependency of summary statistics on model assumptions or parameters are necessary and could be readily assisted by gen3sis.

Studying multiple patterns is a promising approach for disentangling competing hypotheses [20,90]. A wide range of biodiversity dimensions can be simulated with gen3sis (Table 2), which—after appropriate sampling [158]—can be used in a multidimensional comparison with empirical data, i.e., a time series of species abundance matrices and trait matrices, as well as a phylogeny. These output objects are compatible with most R packages used for community or phylogenetic analyses. Hence, the model outputs can be linked to packages computing diversification rates [159], community phylogenetics [160], or functional diversity [161]. The comparison of simulation outputs with empirical data requires a systematic exploration of processes and parameter values when formulating models (e.g., [162]). First, a set of mechanisms and/or a range of reasonable parameter values are explored, e.g., dispersal distances from measurements in a specific clade [163] and/or evolutionary rates [164]. A range of simulation outputs can then be evaluated quantitatively by studying the range of models and parameter values that produce the highest level of agreement with multiple types of empirical data, using, for example, a POM approach [90]. For each model, patterns are evaluated given acceptance criteria (e.g., [42]). A multiscale and multipattern comparison of simulations with empirical data can be completed to evaluate a model's ability to simultaneously reproduce not only one, but a diverse set of empirical patterns across multiple biodiversity dimensions.

Using an illustrative case study, we have demonstrated the flexibility and utility of gen3sis in modeling multiple eco-evolutionary hypotheses in global paleoenvironmental reconstructions (Figs 3 and 4). Our case study indicates that global biodiversity patterns can be modeled realistically by combining paleoenvironmental reconstructions with eco-evolutionary processes, thus moving beyond pattern description to pattern reproduction [35]. Nevertheless, in our case study, we only implemented a few of the standing LDG hypotheses [20,34]. Multiple macroecological and macroevolutionary hypotheses still have to be tested, including the role of stronger biotic interactions in the tropics than in other regions [165], and compared with more empirical biodiversity patterns [20]. Considering multiple additional biodiversity patterns will allow a more robust selection of models. Apart from the global LDG case study, we propose an additional case study (S2 Note, S15 Fig) illustrating how gen3sis can be used for regional and theoretical studies, such as investigations of the effect of island ontology on the temporal dynamics of biodiversity [41,166]. Further, illustrations associated with the programming code are offered as a vignette of the R package, which will support broad application of

gen3sis. Altogether, our examples illustrate the great potential for exploration provided by gen3sis, promising future advances in our understanding of empirical biodiversity patterns.

Gen3sis could be a valuable tool for exploring iconic biodiversity patterns whose underlying mechanisms remain largely under investigation [167]. For example, although we know that mountains are hotspots of biodiversity [56,168], a causal link between mountain dynamics and biodiversity remains poorly understood [169]. Coupling gen3sis with orogenic and erosion models could shed new light on the role of mountain building and associated surface processes in the formation of biodiversity. More generally, the potential role of plate tectonics and surface processes in generating topographic complexity in biodiversity is becoming a hot research topic in the Earth sciences [73]. Similarly, there are many more species associated with coastal reefs (especially coral reefs) in marine ecosystems than in pelagic environments [170]. While it is expected that a combination of geographic features, including plate tectonics [171], and ecological processes interact to form marine diversity, process strengths and interactions are still under investigation [172]. Using gen3sis with paleoenvironmental reconstructions, it is possible to study the interactive effects of ecological and evolutionary processes in shaping global marine biodiversity, with results increasing in precision as more dense and accurate data on paleoenvironmental reconstructions become available [121]. Gen3sis can further support the study of biological processes and can be used to improve our understanding of the links between temperature and biodiversity. For example, it has been hypothesized that temperature influences diversification [63], but the mechanisms and their consequences are still under discussion [173]. Using gen3sis, it is possible to explore the multiple causal pathways between temperature and biodiversity, with the study of the past providing insight into species responses to ongoing climate change [174]. Finally, gen3sis can be used to explore not only species diversity, but also intraspecific genetic structure and thus the correspondence between these diversity levels [175].

## Conclusions

Here, we have introduced gen3sis, a modular simulation engine that enables exploration of the consequences of ecological and evolutionary processes and feedbacks on the emergence of spatiotemporal macro-eco-evolutionary biodiversity dynamics. This modeling approach bears similarity to other computer models that have led to significant progress in other fields, such as climatology [154], cosmology [155], and conservation [156]. We have showcased the versatility and utility of gen3sis by comparing the ability of 3 alternative mechanisms in 2 landscapes to generate the LDG while accounting for other global biodiversity patterns. Besides the LDG, frontiers on the origins of biodiversity involve [16] (i) quantifying speciation, extinction, and dispersal events [119]; (ii) exploring adaptive niche evolution [23,39]; and (iii) investigating multiple diversity dependence and carrying capacity mechanisms [21,115,116]. Further possibilities may include (iv) investigating the mechanisms behind age-dependent speciation and extinction patterns [106,112,176]; (v) exploring contrasts between terrestrial and aquatic ecosystems [16]; and (vi) calculating the uncertainty resulting from climatic and geological dynamics (e.g., [22,23,26,38,43]). Gen3sis can support these research frontiers as a general tool for formalizing and studying existing theories associated with the origin of biodiversity, for testing new hypotheses against data, and for making predictions about future biodiversity trajectories (Table 4). Openly available as an R package, gen3sis has the potential to catalyze interdisciplinary biodiversity research and advance our numerical understanding of biodiversity. We call for the formation of a community of ecologists, biologists, mathematicians, geologists, climatologists, and scientists from other fields around this class of eco-evolutionary simulation models in order to unravel the processes that have shaped Earth's biodiversity.

## Supporting information

**S1 Animation. Reconstructed dynamic landscape L1 (i.e., world 65 Ma) with the environmental values used for the main case study.**
(MP4)

**S2 Animation. Reconstructed dynamic landscape L2 (i.e., world 65 Ma) with the environmental values used for the main case study.**
(MP4)

**S3 Animation. Theoretical dynamic landscape (i.e., theoretical island) with the environmental values used for the supplementary case study.**
(MP4)

**S4 Animation. Dynamic simulated biodiversity patterns (i.e., M5 L1 world from 65 Ma to the present).** The map shows the α diversity and the top and right graphs indicate the richness profile of longitude and latitude, respectively.
(MP4)

**S1 Fig.** Divergence increase per time step $d_i$ against the normalized occupied niche of isolated populations for models (A) M1, M2, M4, and M5, which assume temperature-independent divergence, and (B) M3, which assumes temperature-dependent divergence, where divergence relates to the mean of the realized temperature with 3 different $d_{power}$ values.
(PDF)

**S2 Fig. Nonexhaustive probability density functions of the explored dispersal parameters in a Weibull distribution with shape $\phi$ of 1, 2, and 5 and $\Psi$ of 550, 650, 750, and 850.** Data presented available in S2 Data at https://zenodo.org/record/5006413.
(PDF)

**S3 Fig.** Models (i.e., M1, M2, M3, M4, and M5) (A) Kernel density estimate of the same explored parameters (i.e., divergence threshold and dispersal scale) for selected simulations based on a Pearson correlation of simulated versus best observed (i.e., cor > 0.4) and (B) performance quantified with the BIC. Omitted values from the parameter space were simulations generating an unacceptable best Pearson correlation to the empirical data (r ≤ 0.4), too many species (>35,000) or a weak richness gradient (<20 species between minimal and maximal α-richness). Data presented available in S3 Data at https://zenodo.org/record/5006413. BIC, Bayesian information criteria.
(PDF)

**S4 Fig.** Summary statistics of the model fit to empirical data with and without environmental dynamics for (A) a Pearson correlation of standardized mean species number per latitude (LDG$_{curve}$), (B) a Pearson correlation of spatial α-diversity, and (C) the exact difference between lineage through time curves (nLTT). Data presented available in S2 Data at https://zenodo.org/record/5006413. nLTT, normalized lineage though time.
(PDF)

**S5 Fig.** Standardized mean species number per latitude (LDG$_{curve}$) for empirical data (i.e., terrestrial mammals, birds, amphibians, and reptiles) and best matching simulation from models (A) M1, (B) M2, (C) M3, (D) M4, and (E) M5. Data presented available in S4 Data at https://zenodo.org/record/5006413.
(PDF)

**S6 Fig.** Frequencies of Pearson correlation between simulated standardized mean species number per latitude (LDG$_{curve}$) against best matching empirical LDG$_{curve}$ for each dynamic landscape L1 (in blue) and L2 (in pink) for models (A) M1, (B) M2, (C) M3, (D) M4, and (E) M5. Models M4 and M5 are the only ones producing correlations >0.5. Data presented available in S3 Data at https://zenodo.org/record/5006413.
(PDF)

**S7 Fig. Effects of grid cell size on simulations of M2 L1.** (A) Correlation of grid cell, LDG slope, and other summary statistics. (B) Simulated LDG slope and grid cell size, showing a significant effect of spatial resolution on LDG slope. Data presented available in S5 Data at https://zenodo.org/record/5006413. CPU, central processing unit; LDG, latitudinal diversity gradient.
(PDF)

**S8 Fig. Frequencies of simulated normalized LDG slope (histogram) with empirical LDG for 4 main groups (dashed gray line) and acceptance range (red line).** Frequencies for models (A) M1, (B) M2, (C) M3, (D) M4, and (E)M5 with total frequency and frequency discriminated for each landscape, i.e., L1 and L2. Data presented available in S3 Data at https://zenodo.org/record/5006413. LDG, latitudinal diversity gradient.
(PDF)

**S9 Fig.** Normalized richness of (A) selected simulation, (B) terrestrial mammals, (C) birds, (D) amphibians, and (E) reptiles, with Pearson correlation values for comparisons between simulated and empirical data.
(PDF)

**S10 Fig. Mean absolute evolutionary events (i.e., speciation and extinction) for every 1 myr for the top 7 best matching current spatial α-biodiversity simulations for each model with and without environmental dynamics.** Data presented available in S6 Data at https://zenodo.org/record/5006413.
(PDF)

**S11 Fig. Standardized speciation events for every 1 myr of the top 7 best matching current spatial α-biodiversity simulations for each model with and without environmental dynamics.** Data presented available in S6 Data at https://zenodo.org/record/5006413.
(PDF)

**S12 Fig. Standardized extinction events for every 1 myr of the top 7 best matching current spatial α-biodiversity simulations for each model with and without environmental dynamics.** Data presented available in S6 Data at https://zenodo.org/record/5006413.
(PDF)

**S13 Fig.** Correlation of model parameters and emerging patterns for all models and landscapes without deep-time environmental dynamics (A) M0 L1.0, (B) M0 L2.0, (C) M1 L1.0, (D) M1 L2.0, (E) M2 L1.0, and (F) M2 L2.0. Emerging patterns: (i) phylogeny beta is the phylogenetic tree imbalance statistic measured as the value that maximizes the likelihood in the β-splitting model; (ii) range quant 0.95% is the value of the 95% quantile of the species range area distribution; (iii) LDG % loss is the slope of the linear regression of species richness; (iv) richness r is the highest Pearson correlation between simulated and empirical α-diversity; (v) nLTT diff is the lowest difference between simulated and empirical nLTT curves; and (vi) LDG curve r is the highest Pearson correlation between simulated and empirical standardized mean species number per latitude. Data presented available in S3 Data at https://zenodo.org/

record/5006413. LDG, latitudinal diversity gradient; nLTT, normalized lineage though time.
(PDF)

**S14 Fig.** Correlation of model parameters and 3 emerging patterns for all models and land-scapes considering deep-time environmental dynamics (A) M0 L1, (B) M0 L2, (C) M1 L1, (D) M1 L2, (E) M2 L1, and (F) M2 L2. Emerging patterns: (i) phylogeny beta is the phylogenetic tree imbalance statistic measured as the value that maximizes the likelihood in the β-splitting model; (ii) range quant 0.95% is the value of the 95% quantile of the species range area distribution; (iii) LDG % loss is the slope of the linear regression of species richness; (iv) richness r is the highest Pearson correlation between simulated and empirical α-diversity; (v) nLTT diff is the lowest difference between simulated and empirical nLTT curves; and (vi) LDG curve r is the highest Pearson correlation between simulated and empirical standardized mean species number per latitude. Data presented available in S3 Data at https://zenodo.org/record/5006413. LDG, latitudinal diversity gradient; nLTT, normalized lineage though time.
(PDF)

**S15 Fig.** Results of the island case study showing (A) landscape size and environmental dynamics and (B) results of 3 experiments (i.e., lower, equal, and higher trait evolution compared with the temporal environmental variation). The time series in (B) shows γ richness (log10 scale) on theoretical oceanic islands, following the geomorphological dynamics of islands. Thick lines indicate the average of the replicates, whereas thin lines indicate SD envelopes ($n$ = 30 for each trait evolutionary rate scenario). The dashed gray vertical bar crossing the entire plot indicates the period in which the island reaches its maximum size. Data presented available in S7 Data at https://zenodo.org/record/5006413.
(PDF)

**S1 Note. Global case study: The emergence of the LDG in the Cenozoic.**
(DOCX)

**S2 Note. Island case study: Does trait evolution impact biodiversity dynamics?**
(DOCX)

**S3 Note. Gen3sis pseudo-code.**
(DOCX)

## Acknowledgments

We thank Samuel Bickel and Alex Skeels for thorough comments on this manuscript and package. We thank Camille Albouy, Charles N.D. Santana, Lydian Boschman, Wilhelmine Bach, Thomas Keggin, Flurin Leugger, Victor L.J. Boussange, Conor Waldock, and all sELDiG working group participants for insightful feedback during the model development. We thank the WSL and ETH Zürich for support and infrastructure, including access to High Performance Computing facilities.

## Author Contributions

**Conceptualization:** Oskar Hagen, Loïc Pellissier.

**Data curation:** Oskar Hagen, Fabian Fopp.

**Formal analysis:** Oskar Hagen, Juliano S. Cabral.

**Funding acquisition:** Loïc Pellissier.

**Investigation:** Oskar Hagen, Juliano S. Cabral, Loïc Pellissier.

**Methodology:** Oskar Hagen, Juliano S. Cabral, Florian Hartig, Mikael Pontarp, Loïc Pellissier.

**Project administration:** Oskar Hagen, Loïc Pellissier.

**Software:** Oskar Hagen, Benjamin Flück, Fabian Fopp, Florian Hartig.

**Supervision:** Oskar Hagen, Loïc Pellissier.

**Validation:** Oskar Hagen.

**Visualization:** Oskar Hagen, Juliano S. Cabral, Loïc Pellissier.

**Writing – original draft:** Oskar Hagen, Loïc Pellissier.

**Writing – review & editing:** Oskar Hagen, Juliano S. Cabral, Florian Hartig, Mikael Pontarp, Thiago F. Rangel, Loïc Pellissier.

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
