## [Editor Report · Decision Letter 0]

15 Jul 2020

Dear Dr Hagen, 

Thank you for submitting your manuscript entitled "gen3sis: The GENeral Engine for
Eco-Evolutionary SImulationS on the origins of biodiversity" for consideration as a
Methods and Resources paper by PLOS Biology.

Your manuscript has now been evaluated by the PLOS Biology editorial staff, as well
as by an academic editor with relevant expertise, and I'm writing to let you know
that we would like to send your submission out for external peer review.

Please re-submit your manuscript within two working days, i.e. by Jul 17 2020
11:59PM.

Kind regards,

Roli Roberts

Senior Editor

PLOS Biology

---

## [Decision Letter · Decision Letter 1]

14 Sep 2020

Dear Dr Hagen,

Thank you very much for submitting your manuscript "gen3sis: The GENeral Engine for
Eco-Evolutionary SImulationS on the origins of biodiversity" for consideration as a
Methods and Resources paper at PLOS Biology. Your manuscript has been evaluated by
the PLOS Biology editors, an Academic Editor with relevant expertise, and by three
independent reviewers.

You'll see that while the reviewers recognise the potential merits of your approach,
they are all struggling to identify the clear advance or unique utility that gen3sis
represents over previous initiatives. While the open availability of your platform
is a bonus, the reviewers are currently not fully persuaded by the examples
presented or by your claims to be able to meet the stipulated challenges.

The reviews of your manuscript are appended below. Based on their specific comments
and following discussion with the academic editor, I regret that we cannot accept
the current version of the manuscript for publication. We remain interested in your
study and we would be willing to consider resubmission of a comprehensively revised
version that thoroughly addresses all the reviewers' comments. Note that one
reviewer says "if space is an issue"; at PLOS Biology we have no space restrictions,
but the manuscript should remain well structured and accessible. We cannot make any
decision about publication until we have seen the revised manuscript and your
response to the reviewers' comments. Your revised manuscript would be sent for
further evaluation by the reviewers.

We appreciate that these requests represent a great deal of extra work, and we are
willing to relax our standard revision time to allow you six months to revise your
manuscript. We expect to receive your revised manuscript within 6 months.

**IMPORTANT - SUBMITTING YOUR REVISION**

*Resubmission Checklist*

*Published Peer Review*

*PLOS Data Policy*

*Blot and Gel Data Policy*

Sincerely,

Roli Roberts

Senior Editor,

rroberts@plos.org,

PLOS Biology

REVIEWERS' COMMENTS:

Reviewer #1:

Hagen and colleagues present a tool to simulate eco-evolutionary processes. The
simulation framework can be used to generate null models of biodiversity, and to
test macroecological and macroevolutionary processes across spatio-temporal scales.
The framework is flexible, allowing users to specify various parameters. The authors
suggest the platform represents a much-needed step forward in understanding the
processes that shape biodiversity over Earth history. Similar simulations have been
used to test potential processes that influence biodiversity, but these frameworks
are not readily available for the broader community. Thus, this tool will be of
broad interest but is not novel, and there are now numerous studies that have
employed similar frameworks (see below). In general, I think this is a worthwhile
contribution. However, there are a number of issues that need to be addressed prior
to publication. Chief among them is a lack of clarity regarding the parameters and
functionality of the simulation framework. More detail needs to be provided on
virtually all aspects of the simulation framework, outlined below. The authors also
provide two examples of implementation of the simulation framework. These examples
seem half-baked, without clear hypotheses, null models, or explanation of how the
simulations were performed. The authors need to spend more time discussing both
examples, or, if space is an issue, eliminate one example so that the other can be
given due diligence. Both of these are essential changes. 

1. One of the benefits of a simulation framework is the ability to test competing
processes and hypotheses, and to generate null models. The authors do not adequately
formulate hypotheses for how the processes in the model affect biodiversity
patterns, nor do they implement null models. For example, in the abstract they
indicate that: "Via the process of niche evolution in response to changes in climate
and the spatial distribution of habitat, our model predicts realistic island
biodiversity dynamics, as well as a global latitudinal biodiversity gradient and
species richness distribution". However, it is not at all clear what this means, and
how niche evolution does or does not affect biodiversity patterns. Is niche
evolution necessary to generate realistic biodiversity patterns? How does niche
evolution produce the patterns, and which processes in the simulation (e.g.,
species, dispersal, extinction, carrying capacity) contribute most to the patterns
they find? At what rate do niches need to evolve? These questions remain unclarified
in the body of the text.

2. The case studies were useful for assessing how the simulations can be applied.
However, similar to my comment above, it was not at all clear what the reader should
take away from these exercises. In other words, I was left wondering what the
authors learned from the modelling efforts: which processes were important in
generating the realistic biodiversity patterns? Was it the process of speciation?
Extinction? Dispersal? Again, the benefit of a simulation framework is the ability
to test, explicitly, the relative contribution of processes to the generation of
biodiversity, and to examine what type of patterns result if those processes are not
employed (i.e., a null model). If space is a limitation, the authors should focus on
only one case study and better explain both the methods and results to readers. In
short, I was disappointed in the examples and poor explanation provided by the
authors, especially since the authors suggest that these types of simulations can
provide 'general rules' for the generation of biodiversity, and yet they do not
focus on what they found regarding 'rules' of biodiversity, nor do they employ null
models to ensure results are unique to specific processes. 

3. Line 79: One of the unique aspects provided by the authors' simulation framework
is the ability to assess the effect size of certain variables on the generation of
realistic biodiversity patterns. This can be done by gradually increasing model
complexity, and by implementation of null models. The authors should discuss this
benefit in more detail (around line 79) and employ this framework in their example
case studies. 

4. The authors note that 'practical implementations of simulations have yet to be
consolidated or widely used' (line 74). This is not strictly accurate, and the
authors should point readers to notable exceptions, such as those below (not an
exhaustive list): 

Rangel, T. F. et al (2018). Modeling the ecology and evolution of biodiversity:
Biogeographical cradles, museums, and graves. Science, 361(6399).

Rangel, T. F. et al (2007). Species richness and evolutionary niche dynamics: a
spatial pattern-oriented simulation experiment. The American Naturalist, 170(4),
602-616.

Saupe, E. E. et al (2019). Spatio-temporal climate change contributes to latitudinal
diversity gradients. Nature ecology & evolution, 3(10), 1419-1429.

Saupe, E. E. et al (2020). Extinction intensity during Ordovician and Cenozoic
glaciations explained by cooling and palaeogeography. Nature Geoscience, 13(1),
65-70.

Saupe, E. E. et al (2019). Non‐random latitudinal gradients in range size and niche
breadth predicted by spatial patterns of climate. Global Ecology and Biogeography,
28(7), 928-942.

5. Line 82 and 90: Gotelli and colleagues and Saupe and colleagues have noted that
simulations can be used to examine how continental configuration and paleogeography
influence biodiversity dynamics, which also provide examples of simulations on
million-year time frames (e.g., Saupe et al. 2020. Nature Geoscience). 

6. The authors should provide clarity on the process of extinction in the simulation
framework. When does extinction occur, and can extinction criteria be modified by
the user?

7. The authors refer to 'barriers' within the manuscript, but it was unclear whether
such barriers can be specified, and how they could be specified. I.e., could a
separate input file representing rivers, mountain ranges, etc be used to limit
dispersal? Or, is climate the only potential barrier? Similarly, the authors
indicate that oceanic pixels/cells are inhabitable for simulations focused on the
terrestrial realm: this is very surprising, and I would like the authors to justify
this choice (see line 363). Is this the case for the simulations in general, or can
this feature be 'turned off'?

8. The simulation framework allows for trait divergence and trait evolution. This is
excellent, but the authors should specify what types of traits can be modelled
explicitly in the manuscript. 

9. The links provided in the manuscript to the code are broken. 

10. The authors generally describe the functions in the simulation framework, but
each could benefit from more detail on options and functionality.

11. The process of speciation in the simulation framework was not clear. The authors
indicate that clusters/populations gradually accumulate incompatibility, but it was
not clear at what rate this occurs (line 223). Can the rate be altered? Is the rate
affected by the rate of climate change? The description regarding trait divergence
that eventually leads to speciation differs from the description of divergence on
line 360, which instead refers to numbers of years for speciation to occur (e.g., 2,
4, 6, 8, 10 Ma). Can the authors clarify?

12. Conceptual island case study example (line 299): as noted before, this example
could benefit from additional explanation. It was difficult to understand the
authors' hypotheses/expectations, and their methodology. Did the authors perform
simulations to see what processes should be tweaked to generate a flat biodiversity
gradient on the island? What amount of trait evolution, and which ecological
factors, interact to produce richness gradients?

~12.1: the authors note (line 305) that dispersal was limited to surrounding sites.
Does this mean that populations could only disperse to surrounding cells? If so,
does it follow that speciation could only occur if local extinction happened, which
then isolated populations? 

~12.1.1: Did the authors consider other dispersal thresholds? 

~12.2: What does 'high' and 'low' trait evolution mean? 

~12.3: I was surprised to see that diversity did not decrease when the hypothetical
island decreased in size by erosional processes, following the species-area
relationship. Do the authors have an explanation for why diversity did not decrease,
which seems incongruent with empirical biodiversity patterns? 

13. Latitudinal biodiversity gradient case study example: Again, it was not at all
clear what were the hypotheses for this case study. Which processes did the authors
expect to generate latitudinal biodiversity gradients? Speciation? Extinction?
Carrying capacity alone?

~13.1: Saupe et al in NEE used simulations to evaluate processes that could generate
latitudinal biodiversity gradients. From what I can see, the simulations are similar
to those performed by Saupe and colleagues, although Saupe et al invoked fewer
biotic processes (and therefore their simulations are potentially less realistic).
This contribution seems relevant to cite here (line 419), in addition to reference
19, and should also be discussed more thoroughly throughout this case study. How do
the findings and patterns of the authors compare to previously published simulations
that used a similar simulation framework? Are the processes invoked similar? The
same comment could also be made for the study by Rangel et al 2019 (Science),
although this study was not global in scale: how do the findings compare to the
authors' simulations, which also focused on simulating biodiversity gradients?

~13.2: The authors note they initiated the simulations from a single ancestor species
over the terrestrial surface (line 347). The wording here is unclear, as is the
authors' methodology. Does this imply that the simulation started from only one grid
cell on Earth (if so, which?), or that random grid cells were occupied globally (if
so, which?), but that they all belonged to the same species? 

~13.3: The authors note on line 382 that 'LDG emerged systematically from the
underlying modelled mechanisms', but how? Why? From what mechanisms?

~13.4: It wasn't clear how speciation or extinction was parameterized in these
simulations. Please clarify. 

~13.5: The authors assume a carrying capacity. This is interesting and likely
realistic, but won't this automatically generate more diversity at low latitudes
based on the authors' assumption that carrying capacity is predicated on energy?
This does not seem a true test of process, as the scientific community already knows
that energy is higher at low latitudes. Thus, if you assume more energy allows for a
higher carrying capacity, it will automatically allow a latitudinal biodiversity
gradient to form. What would happen if the authors do not assume a carrying
capacity: is a latitudinal biodiversity gradient produced from the simulations? And,
what differed in the simulations that did not produce biodiversity gradients, which
the authors reference occurred around 10 percent of the time? I'd like to see more
interrogation of these opposing patterns. 

~13.6: The authors note that an increase in K correlated positively with LDG slope,
which is not at all surprising. My question is: what other processes were involved?
Speciation, extinction, dispersal? What process was dominant - i.e., has the largest
effect size?

~13.7: Apologies if I missed this, but I did not see where the authors obtained their
empirical richness patterns, from which to compare to their simulations (line 385). 

~13.8: The authors do not indicate how patterns change when parameters are tweaked.
Are patterns similar at different thresholds (i.e., they altered scale of dispersal,
rates of evolution, speciation time, etc)? What is the sensitivity of biodiversity
patterns to model parameterization? 

~13.9: The climatic data used to power the latitudinal biodiversity gradient example
are highly suspect (detailed in note 2S). There are many weaknesses to what the
authors have done, although they have clearly put a lot of effort into the
reconstructions. The key thing here is that the Koppen reconstructions are
qualitative at best, with large uncertainties associated with them. A few (of many)
criticisms would be:

* They are built from the distribution of sediments, but these have large climatic
ranges associated with them. For instance, bauxites show that it is seasonally wet
(but how seasonal is poorly defined) and warm (above ~22C) (see Price, G., P. Valdes
and B. Sellwood, Prediction of modern bauxite occurrence: Implications for climate
reconstruction. Palaeogeography Palaeoclimatology Palaeoecology, 1997. 131(1-2): p.
1-13.). The temperature could be anywhere between 22C or 32C, or indeed more! This
impreciseness of the indicator is true for pretty much all lithologies used. Hence,
the Koppen maps are a broad-brush estimate of the climate (and this is ignoring the
fact that, for some time periods, there are large data gaps so the maps will be
spatially crude as well). 

* Koppen uses very broad categories. For instance, temperate climates are defined as
anything between 0 and 18C! This is not very precise. As far as I understand, the
authors perform a strange smoothing process (this needs to be clarified) to change
these categories into something quantitative, but you could pretty much do anything
here. They don't seem to think about basic climate phenomena (such as
continentality), so the resulting temperatures will not be physically realistic.
Other Koppen categories are mainly moisture-based, but the same comments apply.

* There is a recent paper that indicates temperature using present day lapse rates is
extremely inaccurate. 

* Interpolating to 170 kyr is spurious. The data is at stage level (~5Ma) and
anything finer has no scientific input.

* Overall, I would be unconvinced that the reconstructions are quantitatively
reliable. At best, they may reflect some very large scale (continental size)
gradients, but I would have no confidence on smaller scales.

* Minor note: reference 3 and 4 seem to be the same in note S2.

14. Line 407: should read as 'flexibility'

15. The authors should examine the example code associated with the 'gen3sis' R
package. The resulting map of alpha richness does not match empirical patterns very
well, and the authors may want to consider a new example. 

Reviewer #2:

This study presents a new R package for conducting simulations of eco-evolutionary
dynamics to explore biodiversity patterns. The authors argue that a reliance on
statistical and verbal models lacking mechanism has hindered progress in
biogeography and macro-ecology and that computer simulations, in which these
mechanisms can be encoded, can provide new insights into the causes of variation in
biodiversity over time and space. The paper describes a general simulation model
aimed to address this, that includes various processes such as dispersal, allopatric
speciation, trait evolution and extinction, that take place on a dynamic landscape.
The paper is well explained, the structure of the model is simple to follow and
overall seems sensible. Two case studies are provided to illustrate the
applicability of the model. The first, demonstrates that when island area rises and
falls over time, this leads to a hump shaped trend in species richness, that is
sensitive to how rapidly niche related traits evolve. The second, shows that when
warm-wet environments have a higher individual carrying capacity, this leads to a
latitudinal gradient in species richness. 

I agree with the authors that these kinds of mechanistic simulation models are an
additional useful tool in understanding biodiversity patterns. Researchers have been
advocating the use of such models for many years and there have been a number of
high profile studies implementing this kind of approach. I have to admit, however,
that I struggled to identify in this paper what the key advance or new insight is
over these earlier mechanistic modelling studies.

While as a modeller, I remain hopeful and open-minded, I am also sceptical that the
general approach advocated here, will address the challenges to the field that the
authors highlight. The problem, is that with these increasingly complex models the
number of free parameters or hard encoded processes becomes enormous. Often the
models are too complex to adequately explore parameter space and to fit to empirical
data. As a result, earlier mechanistic modelling studies have resorting to picking a
particular set of model assumptions and simply showing that this can produce
realistic looking patterns but without formal hypothesis testing or parameter
estimation. For instance, in the current study, the authors show that when warm-wet
environments support more individuals then a latitudinal diversity gradient emerges.
This is expected and of course not a new finding (e.g. Hubbell's 2001 neutral theory
shows that metacommunities with more individuals contain more species), but more
importantly, it doesn't provide much additional insight into whether this mechanism
actually contributes to causing the LDG. 

The problem comes back to the fact that there are multiple explanations for why
richness varies with gradients in temperature and water availability and these
causal hypotheses are extremely difficult to test because they all lead to the same
prediction: that richness is highest in warm-wet environments. If you look at
disciplines such as climate science, which the authors point to as an example of
where mechanistic models have been extremely successful, the reason these complex
models (e.g. GCMs) work is because we have a very good understanding of the basic
physics and chemistry. Without a more thorough understanding of the mechanisms
involved in generating and maintaining biodiversity, and without independent
parameter estimates for most of the key processes, it is difficult to see this
approach leading to major advances. This is a general criticism of the approach, and
one not limited to this specific study, but I failed to see how the authors propose
to address this problem. 

I think the R package that the authors have developed would be a useful tool that I
am sure ecologists would employ. Whether it really provides a 'general' model as the
authors claim I'm not so sure. I think every ecologist will have different opinions
about how a particular process should be modelled, or requirements for what should
or should not be included in the model. For example, I was not convinced by the
first case study on island dynamics presented in the paper, showing that with faster
rates of niche evolution, species diversity is expected to decrease. This arises in
the model because species that evolve rapidly may evolve to have niches that do not
correspond to climate conditions on the island. But it is hard to see how this would
actually happen in nature and it seems highly unrealistic: if the climate affects
individual survival, then species would adapt to the conditions they encounter on
the island and not just randomly drift through niche space. With faster rates of
niche evolution species would be able to better track changing climate conditions
and thus avoid extinction, leading to higher diversity. 

One suggestion, would be for the authors to expand the scope of their experimental
simulations. For instance, to simulate the evolution of biodiversity patterns under
a particular set of model assumptions and parameters (stage 1). Then to employ a
formal model fitting exercise where they run multiple simulations under different
assumptions and parameter combinations and test the fit of these different scenarios
to the patterns produced in stage 1. Testing how often the true model is recovered
(type 1 and type 2 errors), how accurately and reliably the true parameter values
can be estimated, and how these inferences depend on different assumptions would be
an important first step to demonstrating the model is potentially useful. In the
context of the current study, if the authors could demonstrate that richness
correlates with energy-water availability because of differences in individual
carrying capacity, rather than other prominent explanations (e.g. niche
conservatism, niche partitioning etc) then this would be much more compelling. 

Reviewer #3:

This is an interesting and well-written presentation of a new r package for
ecological modelling. The goal of the paper is worthwhile - to create a general
modular software framework for deeper integration of models in geographical ecology
to improve theory and make it more explicit. 

It's a little bit hard for me to tell whether the authors actually reach this worthy
goal. They argue that fields like cosmology and climatology are heavily reliant on
mechanistic models, which they are, but those models seem very different. In
cosmology everything is highly deterministic and fully specifiable once you know the
initial conditions, a modeller's dream. And climate modelling uses a small number of
huge models, greatly dwarfing ecological parallels such as the Madingley model and
clearly completely different from what is offered here. The premise that cosmology
and climate uses models and so should ecology thus seems superficial and in my
opinion doesn't add theoretical depth.

My main question is what this model is for. Is it a toy program that can be used for
teaching? Is it a fully finished model? The authors cast it as a "modelling engine
with a modular implementation", indicating that this is something that other people
can hook up to their code, but how exactly they would do that is not clear. Is it
really a module that can be tweaked and changed and incorporated into other people's
research projects? Or really just a very customisable final model? There's a mention
of the flexibility coming from functions being definable in an open way, but no
example of this (that I can see). As an editorial comment, I don't think the code
examples should be relegated to the supplement in a paper like this - let's see the
package in use in the main text.

This question is especially pertinent given that, as they state, "a mechanstic
understanding" in ecology "is elusive". Do we really know enough about the
mechanisms to say that the implementation choices taken here are sensible? The
authors state in the abstract that the model produces "realistic island biodiversity
dynamics", but how do we know that the emerging dynamics are realistic? How much of
the model framework is amenable to modification, and what elements are taken as
assumed? I can't see that distinction made. 

I have a few other implementation-based questions. It appears that "runtime critical"
elements are implemented via RCpp, but looking at the github code, it's actually a
fairly small part of the code that is written in C++. Is that really enough to make
the model run fast? There are no runtimes given for the examples, which I guess is
fair enough given the diversity of hardware, but would be nice to get an indication
of how fast it is and could be made. I'm here assuming that the code does not call
into any C++ libraries not included in the repo (it doesn't look like it).

Finally it would be nice to see some more expanded documentation for using the
package. There are vignettes available, but are they sufficient to run the software
for other people?

In conclusion, this is potentially interesting, but I don't at present feel I have
sufficient information to evaluate whether the authors achieve what they promise
with the package.

---

## [Decision Letter · Decision Letter 2]

10 Feb 2021

Dear Dr Hagen,

Thank you very much for submitting a revised version of your manuscript "gen3sis: the
general engine for eco-evolutionary simulations on the origins of biodiversity" for
consideration as a Methods and Resources paper at PLOS Biology. This revised version
of your manuscript has been evaluated by the PLOS Biology editors, the Academic
Editor and the original reviewers.

You'll see that while reviewer #3 is now satisfied, reviewers #1 and #2 continue to
raise some significant concerns with your treatment of the literature (and
specifically the novelty of your approach with respect to other studies, such as
Rangel et al, Saupe et al), with the novelty and strength of your LDG case study,
including the parameters and scenarios explored. The Academic Editor asked me to
emphasise the need for you to better explain how the resource builds on existing
work in the field.

In light of the reviews (below), we will not be able to accept the current version of
the manuscript, but we would welcome re-submission of a much-revised version that
takes into account the reviewers' comments. We cannot make any decision about
publication until we have seen the revised manuscript and your response to the
reviewers' comments. Your revised manuscript is also likely to be sent for further
evaluation by the reviewers.

IMPORTANT: I should also say that we will be willing to consult reviewers only once
more on this manuscript, and if they remain unconvinced by the merits of your study
after the next resubmission we would not invite you to revise again.

We expect to receive your revised manuscript within 3 months. 

**IMPORTANT - SUBMITTING YOUR REVISION**

*Re-submission Checklist*

*Published Peer Review*

*PLOS Data Policy*

*Blot and Gel Data Policy*

Sincerely,

Roli Roberts

Senior Editor,

rroberts@plos.org,

PLOS Biology

REVIEWERS' COMMENTS:

Reviewer #1:

In general, I found the manuscript much improved. The R package will be useful to the
evolutionary and ecological communities, although I disagree with the authors
regarding its novelty. To me, the power of the R package is providing an easy-to-use
framework that is flexible with appropriate vignettes explaining the parameters. As
currently written, the authors (perhaps inadvertently) downplay the novelty and
significance of previous, similar platforms.

Most of my questions regarding the framework and its flexibility were clarified by
the authors. They do a nice job explaining the various parameters in Gen3sis, and I
think the algorithm will be of interest to the community. However, and apologies if
I missed this, it was still unclear to me how extinction occurred. I imagine
extinction could occur via numerous routes, and this should be made clear. 

I still found issues with the main LDG case study provided by the authors. Although I
understand the authors provide the case study as an example of how the modelling
framework operates, they also present the results as if the study is novel with
conclusive findings. This is problematic, as the LDG study is not novel (the same
question has been examined using a similar framework by numerous authors) and is
missing key elements, which are discussed below. The authors would have convinced me
of the power of their framework and the use of LDGs as a case study if they had
evaluated more than three hypotheses and included null models. However, this was not
done, even though the authors cite this functionality as one of the primary
novelties of their framework (and it was requested by the reviewers previously). 

1. Hypotheses: 

a. The presentation and description of hypotheses requires further thought. The
authors focus on three hypotheses: time for species accumulation, diversification
rates, and ecological limits. The 'time to speciation' hypothesis is purposely vague
in the literature because it is difficult to disentangle the various processes that
could produce higher species richness at low latitudes using empirical data.
However, a mechanistic model allows for these processes to be examined explicitly.
Speciation, extinction, and dispersal are the only mechanisms that directly generate
differential patterns of biodiversity across landscapes: how might these three
processes contribute to greater species richness in the tropics given an older
tropics, and which are supported by the M0 model? Usually, the 'tropics as older'
hypothesis infers extinction rates were higher at high latitudes due to climate
change (especially in the Northern Hemisphere), but all three could be involved (or
not). 

b. Regarding the carrying capacity hypothesis: the authors should discuss (even
briefly) why higher abundance may lead to higher species richness. I understand that
higher abundance would lead to more biomass, but not necessarily how it could lead
to greater species richness. It is interesting to test this hypothesis in the
Gen3sis framework, but the authors need to better explain the mechanics, as this is
the key contribution of their modelling framework. Again, the only processes that
can produce more species in a region compared to another region are differential
rates of dispersal, speciation or extinction. Thus, how might carrying capacity
generate differential rates of speciation, extinction, or dispersal? Do the authors
suggest that lower carrying capacity will increase extinction of incipient species
at high latitudes? Or, do the authors suggest that speciation and extinction rates
are constant across latitudes, and the higher carrying capacity contributes to
higher diversity in the tropics via dispersal of species into the tropics? Again,
you found this model to be the best supported, and some discussion of how this
occurs in the simulation - the mechanisms - is warranted and would be of great
interest to readers. 

c. Assumptions are provided for each of the models/hypotheses (M0, M1 and M2) in Note
S1, with the exception of Model 0. Model 0 does not seem to be a true null model,
and the assumptions should be specified. Indeed, proper null simulations are not
provided by the authors, which would allow for the effects of certain parameters on
patterns to be examined (e.g., absence of speciation, extinction, ecology). The
authors mentioned null simulations are feasible (for example, in Note S1 'mechanisms
can be completely turned off. In the case of ecology, this will lead to an
ecologically neutral model'), and I was therefore surprised to see they were not
performed. E.g., do flat LDGs result if climate does not change? If extinction or
speciation is prevented, but dispersal allowed? 

2. Results: 

a. How did speciation, extinction, and dispersal contribute to LDG for M2? I found
the better fit of M2 interesting, but I wanted to know more. This is not discussed
by the authors and seems critical to the authors' conclusions. 

b. The figures are very nice. What was interesting, however, was that the simulated
LDG did not fit the empirical LDGs any better (or so it seemed based on Fig S6 and
Fig. 4c) than the LDGs found by Rangel and colleagues and Saupe and colleagues. The
perhaps poorer fit of model M2 to empirical data compared to previous models needs
to be addressed, especially if the authors wish to highlight their conclusions
throughout the manuscript (rather than use the 'case study' as a toy example). 

c. I was particularly intrigued that all models (M0, M1 and M2) resulted in LDGs
(noted by the authors on line 473). This seems an important and interesting finding,
and I would have liked to see more information on how close simulated patterns from
M0 and M1 are to empirical gradients. What are the correlation coefficients between
simulated and empirical curves, and what did the LDG curves look like in these
cases?

3. Discussion: 

a. The authors discuss the congruence of their patterns with Saupe et al, but they do
not compare their results to Rangel et al's seminal work focused on South America,
which analysed the effect of various parameters, including niche evolution (absent
from Saupe et al's model) on LDGs. 

b. I assume the authors could not assess the effect of precipitation on speciation
(found by Saupe and colleagues to be important), as the authors use only temperature
as a variable to constrain niches? 

Minor:

Line 439 and 456: remove apostrophe and replace with comma

Line 101: perhaps not interoperability, but rather implementation of models by
others

Line 498: move the reference to the end of the clause, after 'of the LDG using
computer models"

The references at the end of this sentence also do not examine LDGs sensu stricto,
but do represent mechanistic computer models.

Lines 585-608: I think it would be useful for readers (enhance clarity) if these
lines were moved to the introductory paragraph of the case study. 

Reviewer #2:

I think the authors have gone to a lot of effort to address the comments from the
last review and the paper is improved. I do believe that the R package that the
authors have developed will be a useful tool. However, I'm afraid that I still feel
that the empirical application of the model presented here is far from convincing.
Given the authors claims that these kinds of complex models can provide new insights
into large scale biodiversity patterns, providing a powerful example to demonstrate
this is important. 

The authors now focus on using their model to make inferences about the causes of the
latitudinal diversity gradient. They explore three different scenarios: a time for
speciation model, speciation rate and carrying capacity model. The authors conclude
that the carrying capacity model best fits empirical patterns in terms of matching
the slope of the LDG, the slope of the range size frequency distribution and the
shape of the phylogenetic tree (phylogenetic imbalance). That a model with a
latitudinal gradient in carrying capacity best explain the observed LDG is a strong
conclusion to make but I really don't think the application and fitting of the model
is robust enough to say this.

One concern is the parameter values used. There seemed to be little to no
justification for the choice of parameter values used and as far as I could tell the
authors do not provide evidence showing that their conclusions are robust to a
comprehensive exploration of parameter space. For the parameters they examine, only
a handful of different values are used. I understand that with such a complex model
it is very hard to comprehensively explore parameter space, but I feel this is an
important weakness of the current approach being advocated here. Using slightly
lower or higher values, or indeed some value intermediate to the ones you have
chosen, could change the conclusion of which scenario fits best. Of course, this
possibility can never be ruled out entirely, but I think just trying 3 or 4
different values for a parameter is just nowhere near sufficient. To me the current
implementation of the simulation remains an illustrative, that can be used to make
statements likes 'here are the kinds of patterns that can arise under different
scenarios under these assumptions' but it cannot be used to robustly test
hypotheses. 

The authors only keep simulations that fit certain criteria e.g. fewer than 50,000
species. But it seems like for the scenarios they explored most simulations are
retained. This suggests that a wider range of parameter values needed to be explored
- ideally you want to explore the full breadth of parameter space, so that in one
scenario all species go extinct before the end of the simulation while in another
they all exceed some maximum imposed value. This would tell you that you are
reaching the bounds of realistic parameter values and that there is no need to
explore more extreme values. 

The authors comparison of scenarios also seem incomplete. First, as far as I could
tell from the notes the authors compare a model where there is a latitudinal
gradient in carrying capacity (M2) to models (m0 and m1) where there is no carrying
capacity. But what about the scenario where there is a carrying capacity but it
doesn't vary across latitude? This scenario is needed in order to demonstrate
whether it is the presence of a carrying capacity or the gradient in carrying
capacity which drives differences in model fit. 

I had a query about m1 and the gradient in speciation rates. As far as I could tell
the gradient in time to speciation is ~ 3 (i.e. 3 times faster at the hottest sites)
regardless of the parameter values chosen (e.g. 4/1.3 = 3.1 and 20/6.7 = 3). Perhaps
I'm missing something here, but in that case a clearer explanation is needed. 

When comparing scenarios, the authors identify which model best matches the empirical
data even though the number of parameters varies between models. Some kind of
penalisation for model complexity seems necessary here. Also, the authors narrow
down parameter space by simulating on a coarse grid of 4 degrees, before then
applying those parameters at a finer resolution. To me this doesn't seem
justifiable. The best parameters for explaining the spatial and phylogenetic
patterns will almost certainly vary with the number of grid cells in the simulation.
This will apply to both the absolute parameter values (e.g. a coarser grid with
fewer cells will lead to lower richness) and relative parameter values (e.g. a
coarser grid with fewer cells will limit the extent if variation in range size among
species and thus tree imbalance). 

Another issue is the choice of statistics used to compare empirical and simulated
patterns. In table 2, the authors present a long list of spatial and phylogenetic
metrics that their model can be used to predict. Why then only use three statistics
and why these three? The choice seemed arbitrary and perhaps not obvious. If the
model makes predictions of richness in each cell why not compare the spatial
patterns rather than just the LDG slope which throws so much information away? Why
look at phylogenetic imbalance? Given that one of the hypotheses the authors are
testing is whether there is a carrying capacity to diversity, the temporal dynamics
of species diversification would have seemed an obvious, indeed necessary, metric to
include. Metrics have been developed for matching empirical and simulated LTT plots
that I think would be more informative (e.g. Janzen et al 2015). Related to this,
none of the metrics capture absolute species richness or range size. So, it could be
that the parameter values that give the best match according to the 3 summary
statistics chosen give implausible levels of richness. The metric of Janzen et al
2015 would help to address this. In short, I would be very sceptical that the
metrics chosen here can robustly discriminate between the three different scenarios.
I would also be concerned about the sensitivity of some of these statistics to other
choices in the model and I don't think the current study presents a strong case for
understanding how these statistics depend on the different assumptions of the model
or the parameter values chosen. For instance, previous studies have shown that there
are many factors that can cause differences in phylogenetic tree imbalance, from the
geographic mode of speciation to assumptions about how species compete. Perhaps in
the simulations the authors explored, observed tree imbalance is best matched by the
model with a varying carrying capacity but change the way you model speciation or
competition and this may no longer be the case. 

Janzen, T., Höhna, S. and Etienne, R.S. (2015), Approximate Bayesian Computation of
diversification rates from molecular phylogenies: introducing a new efficient
summary statistic, the nLTT. Methods Ecol Evol, 6: 566-575. https://doi.org/10.1111/2041-210X.12350

L393 -What is the justification for assuming a globally distributed species? How does
this initial condition influence the relative fit of the models?

L405 - it is not immediately clear how this is representing the 'time for species
accumulation' model as the simulation starts with a single globally distributed
species rather than one restricted to the tropics. I assume this is because at the
start of the simulation most environments on Earth were hot and thus this is the
ancestral niche. if that is the rationale then I think this needs to be spelt out
more clearly. 

L407 - Only temperature considered in site suitability? What about precipitation?

Reviewer #3:

I'm happy with the edits to the manuscript - this is a great paper and package. I'll
add that I did not expect this when I started the review, but the author's comments
and edits really convinced me of the usefulness of the engine presented here.

---

## [Decision Letter · Decision Letter 3]

7 Jun 2021

Dear Dr Hagen,

Thank you for submitting your revised Methods and Resources entitled "gen3sis: the
general engine for eco-evolutionary simulations on the origins of biodiversity" for
publication in PLOS Biology. I have now obtained advice from two of the original
reviewers and have discussed their comments with the Academic Editor. 

Based on the reviews, we will probably accept this manuscript for publication,
provided you satisfactorily address the remaining points raised by the reviewers.
Please also make sure to address the following data and other policy-related
requests.

IMPORTANT: Please address the following:

a) Regarding the title, we wonder whether (especially given the reviewers' comments)
it might be more accurate to say "a general engine" rather than "the"? Also, is it
possible to make the second half of the title more explicit. We suggest "gen3sis: a
general engine for eco-evolutionary simulations of the processes that shape Earth’s
biodiversity" (based on a phrase from your Abstract), but are open to other
suggestions.

b) Please address the remaining points from the reviewers. The Academic Editor asked
me "to emphasise reviewer #1's comments and ask the authors to explore further the
potential to build on this method in the Discussion as it is by no means
conclusive."

c) Please attend to my Data Policy requests below. Essentially we'll need the
numerical values presented in Figs 4BC, S2, S3B, S4ABC, S5ABCDEF, S6ABCDE, S7AB,
S8ABC, S10, S11, S12, S13ABCDEFGHIJ, S14ABCDEFGHIJ, S15AB to be made available in
some way; in addition, the location of the data should be clearly stated in the
respective legends.

We expect to receive your revised manuscript within two weeks. 

*Published Peer Review History*

*Early Version*

Sincerely,

Roli Roberts

Senior Editor,

rroberts@plos.org,

PLOS Biology

DATA POLICY:

Many thanks for supplying the code and data required to reproduce the results in
Github. I can see that the global heatmaps are best generated directly from those
files, but for the simpler structured data in your paper, we do ask for all
numerical values that underlie the Figures to be made available in one of the
following forms:

Regardless of the method selected, please ensure that you provide the individual
numerical values that underlie the summary data displayed in the following figure
panels as they are essential for readers to assess your analysis and to reproduce
it: Figs 4BC, S2, S3B, S4ABC, S5ABCDEF, S6ABCDE, S7AB, S8ABC, S10, S11, S12,
S13ABCDEFGHIJ, S14ABCDEFGHIJ, S15AB. NOTE: the numerical data provided should
include all replicates AND the way in which the plotted mean and errors were derived
(it should not present only the mean/average values).

IMPORTANT: Please also ensure that figure legends in your manuscript include
information on where the underlying data can be found (e.g. the supplementary data
files and/or Github), and ensure your supplemental data file/s has a legend.

DATA NOT SHOWN?

REVIEWERS' COMMENTS:

Reviewer #1:

The authors have done a nice job with their revisions, and I am now mostly satisfied
with the manuscript. The model is well described, and its utility is apparent. I
also appreciate how the authors have made their LDG example just that: an example.
The authors have downplayed the results and discussion of the LDG analyses
throughout, and I think this greatly helps with clarity and alleviates the concerns
that I (and the other Reviewer) had. I still find issues with some of their LDG
work. However, I think these issues become less important if the authors can provide
even more explicit discussion that the analyses are simply illustrating the utility
of their framework, and they are not mean to make broad claims about LDG generation. 

1) I am not convinced by the author's justification for assuming a globally
distributed species and how this affects the initial condition for inference of the
models.

2) What does 'modelled process of environmental filtering' mean? (Abstract)

3) The authors partition existing hypotheses for the LDG into three broad categories:
(i) time for species accumulation and niche conservatism, (ii) variation in
diversification rates, and (iii) variation in ecological limit. Although useful to
categorize the many existing hypotheses for the LDG, the categories devised by the
authors seem inaccurate. For example, time to species accumulation does not need to
invoke niche conservatism. Indeed, the 'tropics as older' hypothesis is primarily
about how climate in the tropics has been stable for longer, which allows more
species to accumulate because of reduced extinction. Niche conservatism can play a
role in this hypothesis, but it does not need to do so. The second hypothesis
(variation in diversification rates) should also include discussion of the tropics
as a cradle vs museum. Rates of speciation and extinction can vary across latitudes
for more reasons than increased metabolic and mutation rates. Finally, the third
category invoked by the authors (variation in ecological limit) is also, ultimately,
about reduced extinction due to differing ecological limits. This section needs to
be made clearer, or - alternatively - the authors should focus solely on their
subset of hypotheses, without making claims to the broader set of existing LDG
hypotheses. 

4) I am still concerned about the authors' protocol of removing simulations if they
do not meet certain criteria. It is easy to force simulations to match empirical
data if you only require the simulations to do so some of the time. This, however,
does not necessarily reveal anything deep about underlying mechanisms. For example,
the authors throw out any simulations that do not match the following acceptance
criteria: (i) LDG between 5.4% and 1.1%, (ii) tree shape statistic, β between -1.4
and -0.3, and (iii) range size frequencies with a decrease in the number of
large-range species with a tolerance of 5% (Line 533). 

5) One of the best models found was M4, which imposed a carrying capacity. However,
this carrying capacity did not vary by latitude. I was curious, then, what caused
the LDG to emerge in M4? Do the authors have insight into this?

6) The authors indicate that their results "corroborates with a contribution of both
speciation [22], extinction [6] and dispersal [149] in shaping the LDG" (Line 567).
However, the authors found that dispersal occurs out of the tropics, not into the
tropics (Line 564). Thus, I wondered how dispersal could contribute to the LDG in
this case? Dispersal in this instance seems to be occurring in the reverse direction
to elevate diversity in the tropics. 

Line 252: population instead of populations?

Line 485: distribution, not distributions?

Line 580: South America, not South American 

Line 596: Consider rewording to "we made sure to select parameters based on a range
supported by the literature"

Line 648-651: consider rewording, as this sentence is a bit confusing. 

Reviewer #2:

While I remain skeptical about the utility of these very complex models and would
have expected to see a really impressive empirical demonstration first (the method
would then sell itself!), I think that the computer model that the authors present
will be a useful tool for others and will hopefully encourage further advances. 

The authors have undoubtedly improved the implementation and analysis of their
simulations examining different explanations for the LDG. They have also toned down
their interpretation of these results, admitting that these should be viewed as a
preliminary exploration. So, I think this is now much more balanced and robust.

In short, the authors have addressed my comments.

---

## [Editor Report · Decision Letter 4]

23 Jun 2021

Dear Dr Hagen,

On behalf of my colleagues and the Academic Editor, Andrew Tanentzap, I'm pleased to
say that we can in principle offer to publish your Methods and Resources paper,
"gen3sis: a general engine for eco-evolutionary simulations of the processes that
shape Earth’s biodiversity" in PLOS Biology, provided you address any remaining
formatting and reporting issues. These will be detailed in an email that will follow
this letter and that you will usually receive within 2-3 business days, during which
time no action is required from you. Please note that we will not be able to
formally accept your manuscript and schedule it for publication until you have made
the required changes.

IMPORTANT: Many thanks for providing the underlying data in the Zenodo deposition.
However, please could you include the Zenodo URL in each relevant Figure legend, so
that the Figures are standalone? e.g. in the legend for Fig 4, "Data presented here
is available in S1 Data at https://zenodo.org/record/5006413..." I have told my colleagues to
expect this change.

PRESS: We frequently collaborate with press offices. If your institution or
institutions have a press office, please notify them about your upcoming paper at
this point, to enable them to help maximise its impact. If the press office is
planning to promote your findings, we would be grateful if they could coordinate
with biologypress@plos.org. If you have not yet
opted out of the early version process, we ask that you notify us immediately of any
press plans so that we may do so on your behalf.

Sincerely,

Roli Roberts

Roland G Roberts, PhD 

Senior Editor 

PLOS Biology

rroberts@plos.org